# PRMT5 is an actionable therapeutic target in CDK4/6 inhibitor-resistant ER+/RB-deficient breast cancer

Chang-Ching Lin [1], Tsung-Cheng Chang [2,3], Yunguan Wang[4,5], Lei Guo[4], Yunpeng Gao[6], Emmanuel Bikorimana [1], Andrew Lemoff [7], Yisheng V. Fang[1,6], He Zhang[4], Yanfeng Zhang[4,8], Dan Ye[1], Isabel Soria-Bretones[9], Alberto Servetto[1,10], Kyung-min Lee [1,11], Xuemei Luo[7], Joseph J. Otto [7], Hiroaki Akamatsu[1,12], Fabiana Napolitano [1,10], Ram Mani [6], David W. Cescon [9], Lin Xu [4], Yang Xie [4], Joshua T. Mendell [2,3], Ariella B. Hanker [1] & Carlos L. Arteaga [1] ✉

CDK4/6 inhibitors (CDK4/6i) have improved survival of patients with estrogen receptor-positive (ER+) breast cancer. However, patients treated with CDK4/6i eventually develop drug resistance and progress. *RB1* loss-of-function alterations confer resistance to CDK4/6i, but the optimal therapy for these patients is unclear. Through a genome-wide CRISPR screen, we identify protein arginine methyltransferase 5 (PRMT5) as a molecular vulnerability in ER+/*RB1*-knockout breast cancer cells. Inhibition of PRMT5 blocks the G1-to-S transition in the cell cycle independent of RB, leading to growth arrest in *RB1*-knockout cells. Proteomics analysis uncovers fused in sarcoma (FUS) as a downstream effector of PRMT5. Inhibition of PRMT5 results in dissociation of FUS from RNA polymerase II, leading to hyperphosphorylation of serine 2 in RNA polymerase II, intron retention, and subsequent downregulation of proteins involved in DNA synthesis. Furthermore, treatment with the PRMT5 inhibitor pemrametostat and a selective ER degrader fulvestrant synergistically inhibits growth of ER+/RB-deficient cell-derived and patient-derived xenografts. These findings highlight dual ER and PRMT5 blockade as a potential therapeutic strategy to overcome resistance to CDK4/6i in ER+/RB-deficient breast cancer.

Estrogen receptor-positive (ER+) breast cancer is the most common breast cancer subtype and, as such, is the leading cause of death from this disease[1,2]. Recently, the approval and clinical use of CDK4/6 inhibitors (CDK4/6i) in combination with antiestrogen therapy has significantly improved progression-free and overall survival of patients with ER+ metastatic breast cancer (MBC)[3–6]. Despite these advances, virtually all tumors eventually acquire resistance to this therapy, leaving patients with limited therapeutic options. The efficacy of CDK4/6i relies on an intact retinoblastoma protein (RB)/E2F transcription axis. Inhibition of CDK4/6 activity suppresses RB phosphorylation, enabling RB to

couple to E2F transcription factors and block entry into the S-phase of the cell cycle[7]. Several clinical studies have reported a strong association between *RB1* loss-of-function genomic alterations and resistance to CDK4/6i in patients with ER + MBC[8–12]. Recently, the CDK4/6i abemaciclib was approved as adjuvant therapy for high-risk ER+ breast cancer[13]. With the increased use of CDK4/6i as the standard-of-care for ER+ breast cancer, it is anticipated that RB-deficient breast cancer will become a rising patient population in need of novel therapeutic strategies.

Arginine methylation is an ubiquitous and key post-translational modification (PTM) catalyzed by the PRMT family of enzymes. This

family consists of three types of PRMTs, all of which catalyze the formation of ω-N$^G$-monomethyl arginine (MMA). Type I and type II PRMTs are responsible for the formation of ω-N$^G$,N$^G$-asymmetric dimethylarginine (ADMA) and ω-N$^G$,N$^{'G}$-symmetric dimethylarginine (SDMA), respectively, while type III PRMT catalyzes the formation of only MMA[14]. In mammalian cells, PRMT5 is the primary type II PRMT that methylates arginine residues in the RGG/RG motif[15,16]. Functionally, PRMT5-catalyzed SDMA is recognized by Tudor domains in proteins, thus facilitating protein-protein interactions[17]. Emerging evidence has demonstrated that PRMT5 plays an important role in epigenetic regulation, RNA processing, DNA repair, and cell cycle progression[14,18]. Currently, multiple PRMT5 small molecule inhibitors have entered early-phase clinical trials for solid tumors and hematological malignancies (https://clinicaltrials.gov).

In this work, we perform a genome-wide CRISPR screen using ER+/*RB1*-knockout (RBKO) cells and identify protein arginine methyltransferase 5 (PRMT5) as a molecular dependency in these cells. We find that blocking PRMT5 activity halts the G1-to-S cell cycle transition and effectively arrests cell proliferation, even in the absence of RB, which is the canonical regulator of G1 phase arrest. Mechanistically, inhibition of PRMT5 uncouples fused in sarcoma (FUS) from RNA polymerase II (Pol II), resulting in hyperphosphorylation of Pol II Ser2 and intron retention in multiple genes involved in DNA synthesis during the S phase. Finally, treatment with the PRMT5 small molecule inhibitor pemrametostat and the ER degrader fulvestrant synergistically suppresses growth of ER+/RB-deficient xenografts derived from cell lines and patients. These findings suggest a promising therapeutic strategy of dual blockade targeting both ER and PRMT5 for the treatment of ER+/RB-deficient breast cancer.

## Results

### Genome-wide CRISPR screen identifies PRMT5 as a molecular vulnerability in ER+/*RB1*-deficient breast cancer cells

To mimic *RB1* loss-of-function alterations, we used CRISPR-Cas9 to delete *RB1* in CDK4/6i sensitive MCF-7 and T47D breast cancer cells (Supplementary Fig. 1A). MCF-7_RBKO and T47D_RBKO cells were resistant to treatment with the CDK4/6i abemaciclib, palbociclib and ribociclib, exhibiting 10- to 300-fold higher IC$_{50}$ values for these drugs compared to their isogenic parental cells (Supplementary Fig. 1B–E). Next, we performed a genome-wide CRISPR dropout screen using T47D_WT and _RBKO cells (Fig. 1A). We analyzed the deep sequencing data using MAGeCK[19] and stratified gene essentiality with a false discovery rate (FDR) < 0.05 and β-score < −0.5 (Supplementary Data 1) as previously described[20]. The β-score represents the degree of sgRNA depletion or enrichment, with essential genes having a more negative β-score. For example, *CCND1* (β$_{WT}$ = −1.93; β$_{RBKO}$ = −0.23) and *CDK4* (β$_{WT}$ = −1.75; β$_{RBKO}$ = −0.67) were relatively less essential in RBKO compared to WT cells, consistent with the notion that loss of *RB1* uncouples the CDK4/Cyclin D1 complex from E2F-regulated transcription and the G1-to-S transition[21]. Conversely, *CDK2* and *CCNA2*, both involved in S phase progression, were essential in both cell types (Fig. 1B, bottom half for RBKO and left half for WT cells). We also observed enrichment of sgRNAs targeting *CDKN1B*, *PTEN*, *TSC1* and *TSC2* in both cell types, suggesting deletion of these tumor suppressors provides a survival advantage. In contrast, sgRNAs targeting known oncogenic drivers of ER+ breast cancer were depleted in both WT and RBKO cells. These included essential genes like *MYC*, *PIK3CA*, and *AKT1* (T47D cells harbor an activating *PIK3CA* mutation[22]). Further, sgRNAs targeting essential transcription factors that drive ER signaling, such as *FOXA1*, *GATA3*, *MYC*, *SPDEF*, and *ESR1* itself, were evenly depleted in both WT and RBKO cells (Fig. 1B), suggesting that the ERα pathway may still be essential in these cells irrespective of *RB1* status. We next ranked the essential genes in RBKO cells by selecting the top 50 genes whose corresponding sgRNAs were statistically more depleted in RBKO

over WT cells, and subjected them to gene ontology (GO) analysis[23] to investigate whether these hits converge on a defined molecular function. This gene list was enriched for molecules involved in arginine methyltransferase activity, primarily due to significant depletion of sgRNAs targeting *PRMT5* (β$_{WT}$ = −1.51; β$_{RBKO}$ = −2.76) and *CARM1* (β$_{WT}$ = −0.06; β$_{RBKO}$ = −0.97) (Fig. 1B–D). We chose to focused on PRMT5 because 1) PRMT5 was the top-scoring and druggable hit, 2) literature supports a role for PRMT5 in the progression of various cancer types, including breast cancer[24,25], and 3) PRMT5 small molecule inhibitors (PRMT5i) are in clinical development, thus allowing us to test the antitumor effects of PRMT5 pharmacological inhibition.

### Genetic and pharmacological inhibition of PRMT5 suppress growth of ER+/RB-deficient breast cancer cells

To validate whether PRMT5 was essential for survival of ER+/RBKO cells, we depleted *PRMT5* using CRISPR-Cas9 in both WT and RBKO cells of MCF-7 and T47D lines (Fig. 1E). Consistent with the CRISPR screening results, *PRMT5* depletion resulted in statistical growth inhibition in isogenic WT and RBKO of both MCF-7 and T47D cells, except for one of the sgPRMT5 in MCF-7_WT cells (Fig. 1F). Next, we asked if inhibition of the methyltransferase activity of PRMT5 was required to block growth of ER+/RB-deficient breast cancer cells. To test this, we knocked down *PRMT5* using a doxycycline (DOX)-inducible shRNA targeting the 3′UTR of *PRMT5* in T47D_RBKO cells and then rescued the effect of the shRNA by exogenous expression of either WT PRMT5 or enzymatically dead PRMT5_E444Q. Immunoblot analysis of DOX-treated T47D_RBKO cells showed that expression of WT PRMT5 rescued symmetric dimethylarginine (SDMA) levels that had been suppressed upon induction of the *PRMT5* shRNA. In contrast, expression of PRMT5_E444Q failed to rescue SDMA levels (Fig. 2A). Furthermore, the growth inhibition induced by *PRMT5* knockdown was rescued by the expression of WT PRMT5 but not PRMT5_E444Q (Fig. 2B). These results suggested the potential of inhibiting ER+/RB-deficient breast cancer cell growth by pharmacological inhibition of the PRMT5 methyltrasferase activity. Thus, we next examined the effect of pemrametostat, a competitive inhibitor that binds to PRMT5 substrate binding pocket[26], on growth of RB-deficient cells and a patient-derived xenograft-derived organoid (PDxO; HCI-018)[27]. In a concentration-dependent manner, treatment with pemrametostat markedly decreased SDMA levels in RBKO cells (Fig. 2C), indicating that the inhibitor was engaging its molecular target. Consistent with these results, treatment with pemrametostat over a dose range inhibited growth of MCF-7_RBKO and T47D_RBKO cells with IC$_{50}$ ranged from 49.8 to 268.5 nM (Fig. 2D). However, there were no differences between WT and RBKO cells in sensitivity to pharmacological inhibition of PRMT5 (Fig. 2D; Supplementary Fig. 2). Pemrametostat treatment of PDxO HCI-018 also decreased SDMA levels measured by immunohistochemistry (IHC; Fig. 2E) and inhibited up to 85% growth of the organoids in a concentration-dependent manner (Fig. 2F). Treatment of MCF-7_RBKO and T47D_RBKO cells with sub-μM concentrations of the PRMT5i JNJ64619178 and of CAMA1_RBKO and KPL1_RBKO cells with the PRMT5i GSK591[28] also resulted in growth inhibition (Supplementary Fig. 2), suggesting that the results with pemrametostat also apply to other PRMT5 substrate binding inhibitors. Furthermore, we tested whether ER+ breast cancer cells with low RB expression are sensitive to inhibition of PRMT5. We used shRNA to silence expression of *RB1* in HCC1428 and ZR-75-1 ER+ breast cancer cells. In comparison to the control shRNA targeting GFP (shGFP), shRB1-mediated downregulation of RB resulted in resistance to the CDK4/6i palbociclib. Treatment of the RB-low HCC1428 and ZR-75-1 cells with a dose range of pemrametostat resulted in growth inhibition with sub-micromolar IC$_{50}$, similar to that of MCF-7_RBKO and T47D_RBKO cells (Supplementary Fig. 3). Collectively, these results suggest that PRMT5 is an actionable molecular vulnerability in ER+/RB-deficient breast cancer cells.

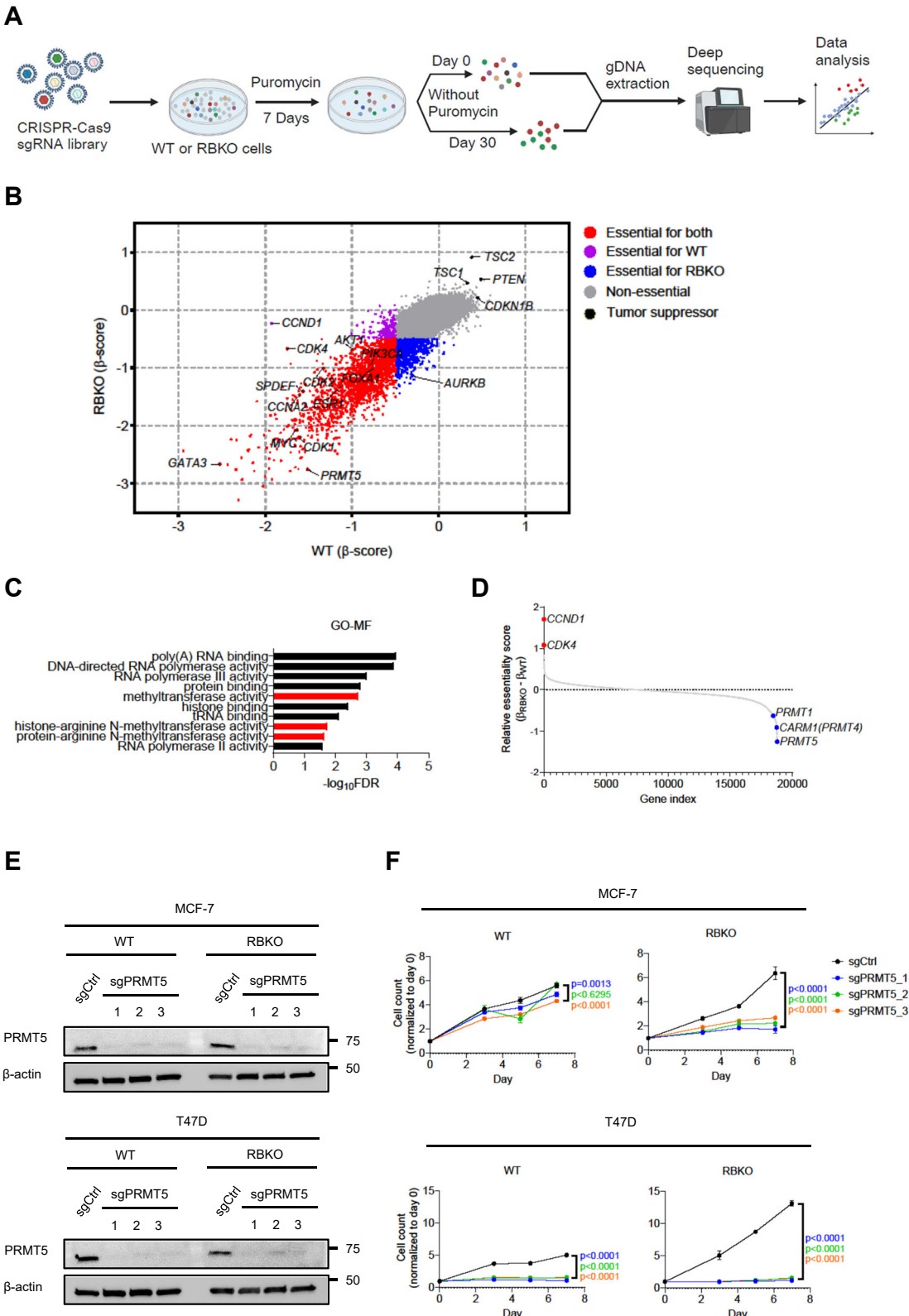

## PRMT5 inhibition blocks G1-to-S cell cycle transition in an RB-independent manner

To investigate the mechanisms underlying the inhibition of cell growth upon silencing of *PRMT5*, we performed RNA-seq on both MCF-7 and T47D cells and compared gene expression of: 1) RBKO vs. WT cells, and 2) RBKO cells transfected with a *PRMT5* siRNA vs. control siRNA. As expected, Gene Set Enrichment Analysis (GSEA) of RNA-seq from RBKO vs. WT cells showed significant upregulation of cell cycle-related Hallmark gene signatures, including E2F targets and G2/M gene signatures in RBKO cells (Supplementary Fig. 4). In contrast, *PRMT5* knockdown in RBKO cells resulted in downregulation of E2F targets, G2/M checkpoint, and mitotic spindle Hallmark gene signatures (Fig. 3A). Furthermore, silencing of *PRMT5* in RBKO cells downregulated 205 genes (in MCF-7) and 473 genes (in T47D) whose

**Fig. 1 | Genome-wide CRISPR dropout screen identifies *PRMT5* as an essential gene for survival of ER + /*RB1*-deficient breast cancer cells. A** Schematic of the CRSIPR screen. Figure was created with BioRender.com. **B** Comparison of the β-scores of the CRISPR screen in T47D_WT and _RBKO cells. The β-score represents the degree of sgRNA depletion or enrichment, with essential genes having a more negative β-score. **C** Gene Ontology-molecular function (GO-MF) analysis using the top 50 genes of which the corresponding sgRNAs were more significantly depleted in T47D_RBKO cells over WT cells. Pathways related to PRMT5 function were highlighted in red. **D** Gene relative essentiality score ranked by the differences of the β-scores in RBKO and WT cells ($\beta_{RBKO} - \beta_{WT}$). Blue and red circles represent genes relatively essential in RBKO and WT cells, respectively. **E** Immunoblot analysis of MCF-7 and T47D cell lysates. Non-targeting control sgRNA (sgCtrl) and three sgRNAs targeting *PRMT5* (sgPRMT5) were individually transduced to the cells (n = 2 biological replicates). Lysates were probed with the indicated antibodies. **F** Monolayer growth of sgCtrl- or sgPRMT5-transduced MCF-7 and T47D WT and RBKO cells. Cells were counted using a Coulter counter. Data represent mean ± SD (n = 3 biological replicates), one-way ANOVA with a Dunnett's post-hoc test, p-values depict color coded group vs sgCtrl. Source data are provided as a Source Data file.

expression had been increased by *RB1* knockout. GSEA of those (205 and 473) genes in both cell lines showed highly statistical enrichment of E2F targets and G2/M checkpoint gene signatures (Fig. 3B), suggesting that silencing of *PRMT5* reversed the changes on cell cycle gene expression induced by *RB1* loss.

Next, we sought to examine whether PRMT5 inhibition resulted in dysregulation of the cell cycle. We employed siRNAs to knockdown *PRMT5* in MCF-7_RBKO and T47D_RBKO cells and then performed cell cycle analysis of propidium iodide-stained cells. *PRMT5* knockdown led to accumulation of cells in G1 phase and reduction of cells in S phase (Fig. 3C, D). Similar to PRMT5 siRNA, treatment with pemrametostat in MCF-7_RBKO cells also hampered the G1-to-S phase transition in a dose-dependent manner (Fig. 3E) and suppressed the expression of E2F target genes as measured by qRT-PCR (Fig. 3F).

Since *RB1* loss-of-function mutations occur across various tumor types, we also examined the effect of *PRMT5* silencing in other RB-deficient cancer cell lines. *PRMT5* knockdown in lung cancer (H596, H1048 and H1155), prostate cancer (Du-145) and triple-negative breast cancer (MDA-MB-436) cell lines, all harboring natural *RB1* loss-of-function mutations or deletion, resulted in significant growth inhibition, accumulation of cells in G1 phase, and a decrease in cells in S phase (Supplementary Fig. 5), further suggesting the potential applicability of therapeutic targeting of PRMT5 across various types of cancer with RB deficiency.

## FUS is a functional substrate of PRMT5

RB plays a pivotal role in the G1-to-S checkpoint during cell cycle progression[29,30]. The effect of PRMT5 inhibition on blocking the G1-to-S transition in *RB1*-deleted cells suggests a potential approach to suppress cancer cell proliferation. Since the methyltransferase activity of PRMT5 was essential to support growth of ER + /RBKO cells (Fig. 2A, B), we aimed to identify downstream effectors of PRMT5. Thus, we employed co-immunoprecipitation (Co-IP) mass spectrometry (MS) analysis and found 192 proteins that were significantly enriched in PRMT5 antibody pulldowns compared to IgG control (Supplementary Data 2). As expected, PRMT5 was one of the most significantly enriched proteins identified by MS. Consistent with previous studies[16,31], we also found significant enrichment of the PRMT5 hetero-octamer partner MEP50 (also known as WDR77) and substrate adaptors COPR5 and pICln (Fig. 4A). To determine putative substrates of PRMT5's enzymatic activity, we performed a SDMA post-translational modification (PTM) analysis using LC-MS/MS to examine SDMA level changes in PRMT5-associated proteins upon *PRMT5* knockdown. The PTM analysis identified 165 SDMA peptides, of which 21 peptides exhibited significant downregulation of SDMA levels when *PRMT5* was knocked down (Fig. 4B; Supplementary Data 3). Integrating the results from the Co-IP MS and the SDMA PTM analysis with PRMT5 substrates reported in published studies[16,31] identified five common hits (FAM120A, FUBP1, FUS, FXR2 and G3BP1) (Fig. 4C). Among these five candidates, *FUS* (fused in sarcoma) was the most essential gene ($\beta_{WT} = 0.17$; $\beta_{RBKO} = -0.61$) in the initial CRISPR screen of ER+/RBKO cells (Fig. 4C; Supplementary Data 1).

FUS is a DNA/RNA-binding protein involved in the regulation of gene transcription, DNA repair, and RNA processing[14]. We confirmed the interaction between PRMT5 and FUS by reciprocal Co-IP of endogenous PRMT5 and of FUS followed by immunoblot analysis with FUS and PRMT5 antibodies, respectively (Fig. 4D). In addition to PRMT5, FUS also interacted with MEP50, suggesting that FUS may be involved in the PRMT5 methylosome (Fig. 4D). We then monitored cell proliferation and performed cell cycle analysis upon knockdown of *FUS* with siRNAs in MCF-7_RBKO and T47D_RBKO cells. Knockdown of *FUS* significantly decreased RBKO cell viability and disrupted their G1-to-S cell cycle transition (Fig. 4E–G), thus phenocopying the effects of *PRMT5* inhibition (Figs. 2 and 3). Taken together, these results were consistent with the proteomics analysis and supported FUS as a functional substrate of PRMT5 in ER+/RBKO cells.

## PRMT5 inhibition results in hyperphosphorylation of Ser2 Pol II and dysregulation of RNA splicing

FUS is known to form a liquid droplet phase-separated structure that mediates bindings with the carboxyl-terminal domain (CTD) in RNA polymerase II (Pol II)[32–34]. Depletion of *FUS* has been reported to derepress Ser2 phosphorylation (pSer2) in the CTD of Pol II, resulting in abnormal accumulation of pSer2 Pol II and dysregulation of gene transcription and RNA splicing[35,36]. Therefore, we speculated that 1) PRMT5-catalyzed SDMA levels in FUS are necessary for the interaction between FUS and Pol II, and 2) PRMT5 inhibition results in dissociation of FUS from Pol II. Supporting this hypothesis, treatment with pemrametostat in MCF-7_RBKO and T47D_RBKO cells significantly suppressed SDMA in FUS and reduced the association of FUS with PRMT5, MEP50, and Pol II, as shown by FUS antibody pulldowns followed by immunoblot analysis (Fig. 4H, I). Next, we asked whether PRMT5 inhibition increased pSer2 Pol II as a result of uncoupling FUS from Pol II. To this end, we investigated the effects of pemrametostat on the distribution of pSer2 Pol II using chromatin immunoprecipitation-sequencing (ChIP-seq) with a pSer2 Pol II-specific antibody. ChIP-seq analysis identified 26,523 common peaks of pSer2 Pol II chromatin binding (FDR < 0.05) in MCF-7_RBKO cells treated with pemrametostat or DMSO. Pemrametostat-mediated inhibition of PRMT5 in MCF-7_RBKO cells resulted in gains of 9,633 unique pSer2 Pol II chromatin binding peaks as opposed to a 6-fold lower loss of 1,507 binding peaks (Fig. 5A, B). This global increase of pSer2 Pol II chromatin binding was in line with a previous study that showed accumulation of pSer2 Pol II upon of loss of FUS[35].

Since the function of PRMT5, FUS, and pSer2 Pol II converges on regulation of RNA splicing[16,35–41], we next performed RNA-seq of MCF-7_RBKO cells treated with pemrametostat to examine changes in RNA splicing. Pemrametostat-mediated inhibition of PRMT5 resulted in significant changes in RNA splicing, particularly in intron retention (IR) events (Fig. 5C). Transcripts with IR are either degraded by nonsense-mediated decay (NMD) or detained in the nucleus (also known as detained intron) and then degraded before protein translation[37,42]. Either NMD or a detained intron will result in lower protein translation. To assess this, we next used IRFinder, an algorithm designed to identify IR events with higher precision and accuracy than MISO and DEXseq[43]. In RNA from cells treated with pemrametostat vs. DMSO, we identified 2,778 significant IR events (FDR < 0.05) in 1,185 genes; 41% (489/1,185) of these genes also gained pSer2 Pol II chromatin binding

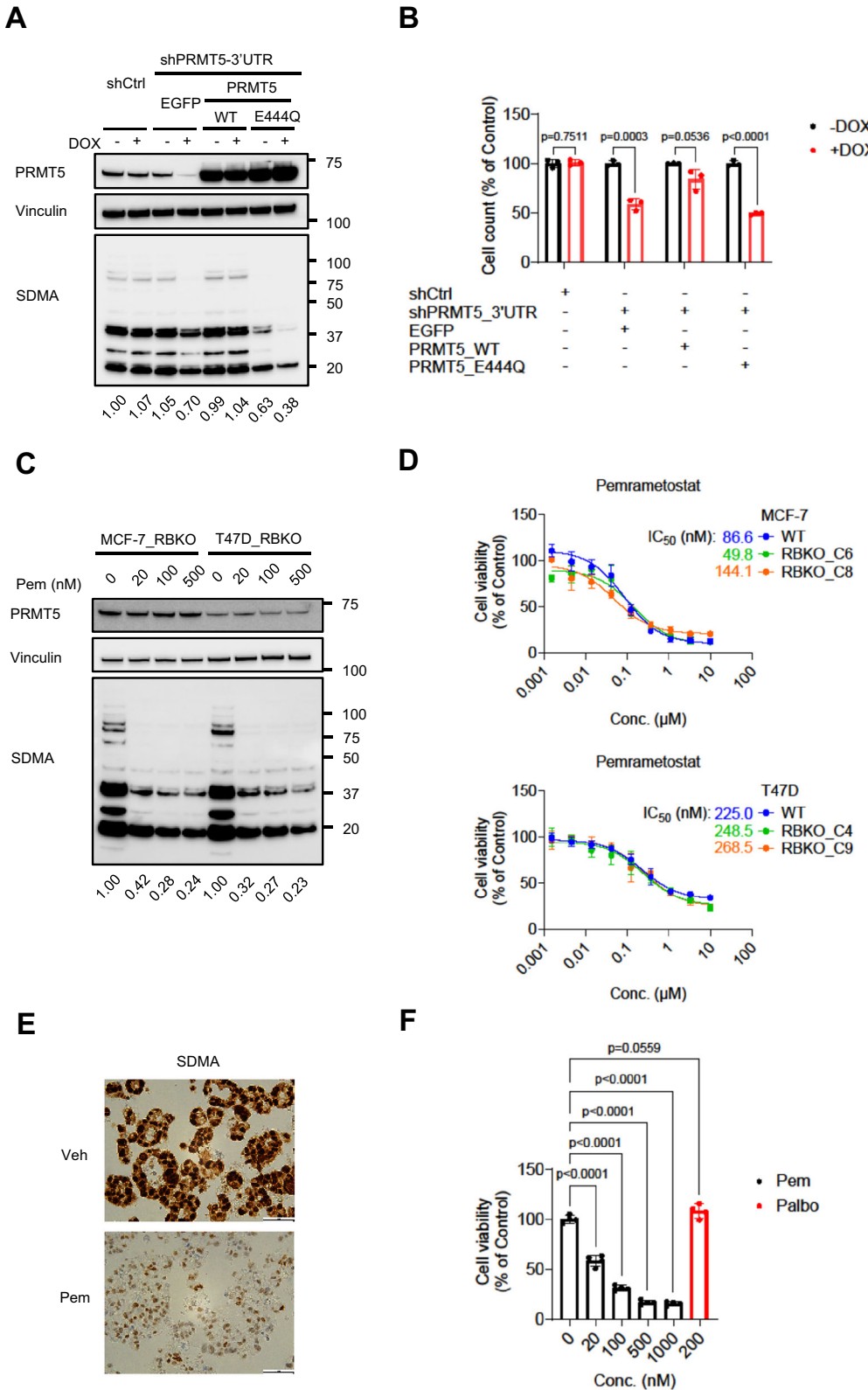

(Fig. 5D), in line with the notion that pSer2 Pol II is associated with regulation of RNA splicing[35,44]. Finally, we performed pathway analysis on these 1,185 genes, stratifying them base on whether they gained (489 genes) or did not gain (696 genes) pSer2 Pol II binding. Among the genes that gained pSer2 Pol II chromatin binding, cell cycle-related pathways were among the top enriched pathways (Fig. 5E), whereas the group of genes without gain in pSer2 Pol II binding was mainly enriched for RNA processing-related pathways (Fig. 5F). This gene-specific

pattern suggests that the increase in pSer2 Pol II resulting from inhibition of PRMT5 is associated with intron retention of genes that regulate cell cycle progression.

The pathway analysis also identified multiple genes with pemrametostat-induced IR that are involved in DNA replication (e.g., *ANAPC7, CDC45, GINS1, LIG1, MCM4, ORC2, POLD2, POLD3, POLE, RFC3,* etc.) (Fig. 5G, H). The majority of these genes had minor changes in gene expression (log$_2$FC within ±0.5). Thus, we next investigated

**Fig. 2 | Targeting PRMT5 inhibits growth of ER+/RBKO breast cancer cells.**
**A** Immunoblot analysis of T47D_RBKO cells transduced with a doxycycline (DOX)-inducible control shRNA (shCtrl) or a shRNA targeting *PRMT5* 3'UTR (shPRMT5-3'UTR). The shPRMT5-3'UTR cells were further transduced with EGFP, WT PRMT5, or PRMT5_E444Q. Cells were treated with or without 200 ng/mL DOX for 4 days; cell lysates were collected and subjected to immunoblot analysis. Numbers in the bottom represent fold changes of SDMA levels (normalized with vinculin, n = 2 biological replicates). **B** Monolayer growth of T47D_RBKO cells as described in (A). Cells were treated with (red) or without (black) 200 ng/mL DOX for 8 days. Cell number was counted using a Coulter counter. Data represent mean ± SD (n = 3 biological replicates), two-sided Student's t test. **C** Immunoblot analysis of MCF-7_ and T47D_RBKO lysates. Cells were treated with pemrametostat (Pem) for 3 days. Lysates were collected and subjected to immunoblot with the indicated antibodies.

Numbers in the bottom represent fold changes of SDMA levels (normalized with vinculin, n = 3 biological replicates). **D** Dose response curves of Pem in WT or RBKO clones. Cells were treated with Pem (0-10 μM) for 6 days. Cell viability was measured by the CyQuant assay. Data represent mean ± SD (n = 3 biological replicates). **E** Immunohistochemistry (IHC) of SDMA. The ER+/*RB1*-deleted organoid HCI-018 was treated with vehicle control (Veh) or 500 nM Pem for 6 days. The organoids were then harvested, fixed, and then embedded for IHC (n = 2 biological replicates). The scale bars on the lower right represent 50 μm. **F** Viability of HCI-018. The organoids were treated with Veh, Pem, or palbociclib (Palbo) for 20 days and then subjected to the 3D CellTiter-Glow assay. Data represent mean ± SD (n = 4 biological replicates), one-way ANOVA with a Dunnett's post-hoc test. Source data are provided as a Source Data file. Source data are provided as a Source Data file.

whether IR in these genes, resulting from PRMT5 inhibition, is associated with a reduction of their corresponding protein levels. Treatment of MCF-7_RBKO and T47D_RBKO cells with pemrametostat resulted in a clear protein downregulation of APC7, GINS1, ORC2, and POLE as measured by immunoblot (Fig. 5I, J), except for POLE in T47D_RBKO cells. These results suggest a causal association between the PRMT5-FUS-Pol II axis and proper RNA splicing of genes involved in DNA replication (Fig. 6). We propose this as a mechanism by which PRMT5 inhibition impedes entry into S phase irrespective of *RB1* status. Supporting this hypothesis, cell cycle analysis also showed accumulation of *RB1*-deleted cells in G1 phase and a decrease in S phase upon PRMT5 genetic and pharmacological inhibition (Fig. 3D, E; Supplementary Fig. 5C).

**Therapeutic inhibition of PRMT5 synergizes with antiestrogens against ER+/RB-deficient breast cancer**
Finally, we investigated effective therapeutic combinations to treat ER+/RB-deficient breast cancer. Our CRISPR screen identified that drivers of ERα signaling remained essential for the viability of ER+ breast cancer cells irrespective of *RB1* status (Fig. 1B), suggesting that ER+/RB-deficient breast cancer cells may still be dependent on ERα signaling. Thus, to test whether antiestrogens can be leveraged for treatment of ER+/RB-deficient breast cancer, we treated MCF-7 and T47D, both WT and RBKO cells with fulvestrant (a selective ERα degrader) or switched them to estrogen-free media to mimic estrogen suppression with aromatase inhibitors as is done in patients. Both treatments with fulvestrant and estrogen deprivation inhibited 60-80% of cell growth, blocked the G1-to-S transition, and reduced ER transcriptional activity, as assayed with an estrogen response element (ERE) luciferase reporter, in both WT and RBKO cells (Supplementary Fig. 6). Addition of 17β-estradiol rescued the inhibitory effects of estrogen deprivation on cell growth, ER transcriptional reporter activity, and cell cycle progression (Supplementary Fig. 6). Since estrogen suppression, fulvestrant, and PRMT5i all exerted blockade of the G1-to-S transition independent of RB, we reasoned that anti-ER therapy plus a PRMT5i may serve as an effective combination strategy to treat ER+/RB-deficient breast cancer. To evaluate the antitumor action of the combination, we treated MCF-7_RBKO and T47D_RBKO cells with fulvestrant and pemrametostat across a dose range and then calculated the combination index using the Chou-Talalay method[45]. After 6 days of treatment, the drug combination resulted in greater growth inhibition of RBKO cells than either fulvestrant or pemrametostat alone, with combination indices ranging between 0.50 to 0.85, suggesting a synergistic effect of the combination (Fig. 7A).

We then tested the effect of dual blockade of ER and PRMT5 in vivo. We treated female nude mice bearing established MCF-7_RBKO xenografts with fulvestrant, pemrametostat, or the combination for 60 days. Treatment with fulvestrant and pemrametostat each alone delayed tumor growth compared to the control arm, whereas the combination of both drugs arrested tumor growth. Out of 9 mice, 6

exhibited partial tumor remission and 1 showed complete remission (Fig. 7B). IHC analysis revealed significant downregulation of ERα and SDMA expression in tumors treated with fulvestrant and pemrametostat, respectively, confirming target inhibition. Moreover, treatment with each inhibitor alone or in combination significantly decreased the number of Ki67 positive cells compared to the control arm (Fig. 7C; Supplementary Fig. 7). Similar results were observed in mice bearing *RB1*-deleted patient-derived xenografts (PDX) derived from a patient with ER+ MBC that progressed clinically on palbociclib plus the aromatase inhibitor letrozole. *RB1* deletion in the PDX was confirmed by exome sequencing (Materials and Methods), and loss of RB protein expression was confirmed by immunoblot analysis and IHC using an RB antibody (Supplementary Fig. 8). Single agent pemrametostat, but not fulvestrant, delayed growth of the *RB1*-deleted PDXs compared to vehicle control. However, the combination of fulvestrant and pemrametostat induced durable tumor suppression over 70 days of treatment, with 4 out of 8 mice exhibiting partial tumor remission (Fig. 7D). Treatment with the combination also suppressed the expression of ERα, SDMA, and Ki67 in the *RB1*-deleted PDXs (Fig. 7E; Supplementary Fig. 7). Collectively, our data suggest that dual blockade of ER and PRMT5 can effectively suppress tumor growth of ER+/RB-deficient breast cancer, thus providing the basis for testing this therapeutic combination in patients with this refractory breast cancer genotype.

## Discussion
In this study using a genome-wide CRISPR screen, we identified PRMT5 as a molecular dependency in ER+/RB-deficient breast cancer. Loss of the tumor suppressor RB is an established mechanism of de novo and acquired resistance to CDK4/6i, and with the wider use of these agents as standard treatment, the population of patients with RB-deficient breast tumors is likely to rapidly increase. To the best of our knowledge, a targeted therapeutic approach against these cancers once they progress on a CDK4/6i has not yet been established. Thus, PRMT5 represents an actionable therapeutic vulnerability in breast cancers of this genotype and potentially fulfills an unmet need for patients with acquired resistance to CDK4/6i.

Inhibition of PRMT5 blocked the G1-to-S cell cycle transition in ER+/RBKO breast cancer cells and other cancer cells with natural loss-of-function alterations of *RB1*. Of relevance to our results, Abu-Hammad et al. recently reported that PRMT5 is an indirect target of CDK4 and is required for the sensitivity of RB-competent melanoma cells to palbociclib[46]. In this study, treatment of melanoma cells with palbociclib resulted in suppression of PRMT5 activity, which altered pre-mRNA splicing of *MDM4* and downregulated MDM4 protein levels. This in turn led to activation of p53, induction of p21 expression, and subsequent inhibition of CDK2, thus providing a mechanism of action of CDK4/6i in RB-proficient cells. Although their study did not focus on RB-deficient cancers, it provided important insights into the mechanism by which PRMT5 regulates cell cycle progression. Our data, however, show that siRNA-mediated silencing of *PRMT5* resulted in suppression of cell growth and entry into S

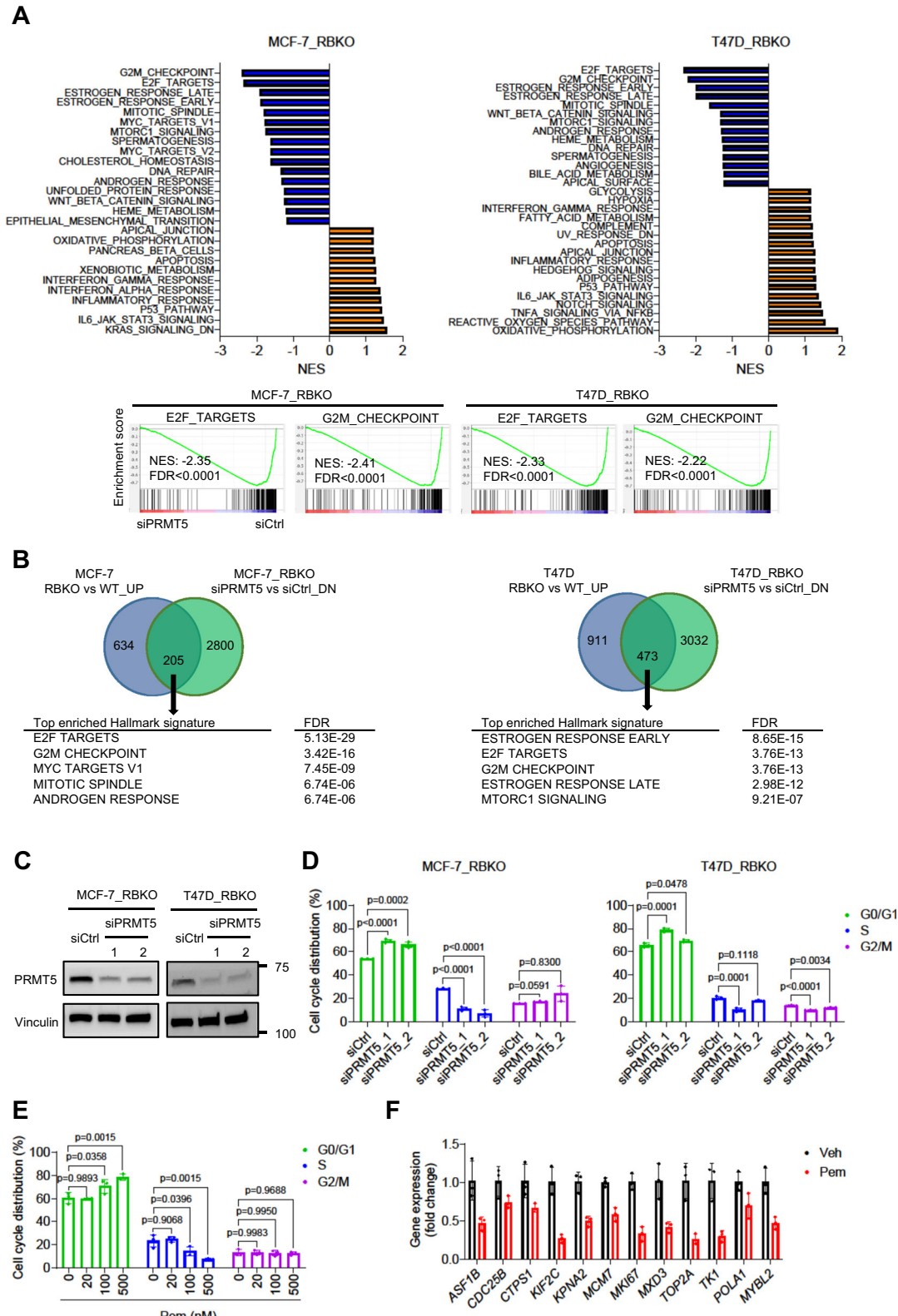

phase in both p53 mutant (T47D) and p53 wild-type (MCF-7) CDK4/6-resistant cells lacking RB. This suggests that, in addition to MDM4-p53, other regulatory axes may also play a role in PRMT5-mediated regulation of the G1-to-S transition independent of RB and CDK4/6.

An unbiased proteomics approach (e.g., Co-IP MS and SDMA PTM analysis) allowed us to identify the DNA/RNA binding protein FUS as a putative substrate of PRMT5. Importantly, *FUS* knockdown phenocopied the effects observed with *PRMT5* silencing in RBKO cells.

Previous studies have reported that silencing of *FUS* results in abnormal accumulation of pSer2 Pol II and RNA splicing defects[35,36]. Consistent with these studies, our ChIP-seq and RNA-seq analysis revealed that treatment with pemrametostat induced an increase in pSer2 Pol II levels associated with intron retention (IR) within genes significantly enriched for cell cycle-related pathways, thus supporting a regulatory role of the PRMT5-FUS-Pol II axis in RNA splicing and cell cycle progression. Notably, FUS itself is essential to bridge the interaction

**Fig. 3 | Silencing of *PRMT5* downregulates E2F gene signature and blocks G1-to-S-phase progression. A** Gene set enrichment analysis (GSEA) of the Hallmark gene signatures. MCF-7_RBKO and T47D_RBKO cells were transfected with control siRNA (siCtrl) or a *PRMT5* siRNA (siPRMT5) for 3 days. Total RNA was extracted from the cells and then subjected to RNA-seq. NES: normalized enrichment score; FDR: false discovery rate. Upregulated and downregulated gene signatures were shown in orange and blue, respectively. **B** Venn diagram showing differentially expressed genes in RBKO vs WT cells and siPRMT5 vs siCtrl in RBKO cells. Cutoff FDR < 0.01 and fold change > 0.2. **C** Immunoblot analysis of MCF-7_RBKO and T47D_RBKO cells. Cell lysates were collected 3 days after transfection of siCtrl or two individual siPRMT5 (n = 2 biological replicates). The lysates were probed with antibodies as indicated. **D, E** Cell cycle analysis. Cells were fixed 3 days after transfection of siCtrl or two individual siPRMT5 (**D**). MCF-7_RBKO cells were treated with vehicle control (Veh) or Pemrametostat (Pem) for 3 days and then were fixed (**E**). The fixed cells were stained with propidium iodide and then subjected to flow cytometry analysis. Data represent mean ± SD (n = 3 biological replicates), one-way ANOVA with a Dunnett's post-hoc test. **F** Expression of E2F target genes in MCF-7_RBKO cells. Cells were treated with Veh or 500 nM Pem for 3 days and then subjected to total RNA extraction, reverse transcription, and qRT-PCR. Data represent mean ± SD (n = 3 biological replicates). Source data are provided as a Source Data file.

between the splicing factor U1 snRNP and Pol II[47]. Since inhibition of PRMT5 uncoupled FUS from Pol II, further investigations are warranted to decipher if U1 snRNP or other splicing factors contribute to RNA splicing of cell cycle regulators. Of note, previous studies have reported that PRMT5 regulates RNA splicing via arginine methylation of splicing factors and subunits of the spliceosome[16,31,38]. Therefore, we acknowledge that inhibition of PRMT5 may suppress cell cycle progression independent of the PRMT5-FUS-Pol II axis. Indeed, our results showed that a portion of the retained introns induced by PRMT5 inhibition is unrelated to pSer2 Pol II. Although these retained introns are enriched in RNA processing pathways, we cannot rule out the possibility that they could indirectly inhibit G1-to-S transition. In addition to RNA splicing factors, FUS is also associated with TFIID transcription complex, suggesting that FUS may play a role in initiation of Pol II-dependent transcription[48]. Furthermore, FUS binds to Pol II on alternative poly-adenylated sites to regulate transcription termination[36]. These studies suggest that dissociation between FUS from Pol II may result in direct dysregulation of gene expression. Therefore, further studies are needed to investigate whether PRMT5i-mediated dissociation of FUS and Pol II leads to mRNA downregulation of cell cycle regulators.

Recent studies have demonstrated compensatory crosstalk between PRMT1 and PRMT5, and combined inhibition of PRMT1 and PRMT5 leads to synergistic antitumor effects[40,49,50]. Indeed, FUS is also a substrate for PRMT1[51]. Although we did not examine levels of MMA and ADMA in FUS or in tumors in vivo, it is possible that other PRMT family proteins can partially compensate pemrametostat-mediated SDMA suppression in FUS. Of note, our CRISPR screen also identified Type I PRMTs (e.g., PRMT1 and CARM1) as essential genes in RBKO cells. Therefore, further investigation for dual inhibition of PRMT1 and PRMT5 is warranted.

The activating cofactor S-adenosyl-l-methionine (SAM) serves as the methyl group donor required for PRMT5's methyltransferase activity. As a result, first-generation PRMT5i under clinical development are either SAM-cooperative or SAM-competitive[52]. Therapeutic inhibition of PRMT5 has been proposed as synthetically lethal in cancers with *MTAP* loss. In *MTAP*-deleted tumors, increased intracellular concentrations of methylthioadenosine (MTA), the metabolite cleaved by the MTAP enzyme, couple with PRMT5, compete with SAM, and inhibit PRMT5's enzymatic activity. This contributes to the suggested synthetic lethality in these cancers[53,54]. Since first-generation PRMT5i do not target the PRMT5-MTA complex, it is not clear yet whether they will be clinically active in patients with MTAP-deficient cancers. It has been proposed that treatment of these tumors may require a selective binder to the PRMT5/MTA complex[55]. In our study herein, we demonstrate that both SAM-cooperative (e.g., pemrametostat) and SAM-competitive (e.g., JNJ-64619178) PRMT5i suppress growth of both ER+/RB-deficient and RB-competent breast cancers in vitro and/or in vivo. Inhibition of PRMT5 induces intron retention and subsequently downregulates corresponding proteins that drive DNA synthesis in the S phase. Therefore, inhibition of PRMT5 bypasses the CDK4/6-RB-E2F regulatory axis and thus impedes G1-to-S transition independent of RB. Although targeting PRMT5 is not synthetic lethal to RB-deficiency, its

mechanism of cell cycle inhibition provides a rationale for future studies testing first-generation PRMT5i also in RB-competent, CDK4/6i-refractory breast cancers with other mechanisms of resistance (e.g., *CCNE1* overexpression, *FAT1* loss, *PTEN* loss, etc.)[9,12,56,57].

Our data also suggest that the ERα pathway may still be essential in ER+ breast cancer cells lacking RB and as such remain sensitive to estrogen suppression and ER antagonists. This finding is in line with a recent study by Wander SA et al., which showed growth of *RB1*-deleted breast cancer cells is still inhibited by fulvestrant[12]. In our study, the combination of PRMT5i and fulvestrant exhibited superior antitumor activity compared to either monotherapy alone against xenografts of this breast cancer genotype. Hence, we propose that the combination of ER and PRMT5 inhibitors can synergistically block the G1-to-S transition in ER+/RB-deficient breast cancer, independent of the CDK4/6/Cyclin D1 complex.

In summary, our results provide evidence that targeting the arginine methyltransferase activity of PRMT5 blocks the G1-to-S cell cycle transition independent of RB. Additionally, we demonstrate the association between the PRMT5-FUS-Pol II axis and intron retention within genes that are enriched for cell cycle progression in ER+/RB-deficient breast cancer cells. Collectively, these data support PRMT5 as a therapeutically actionable vulnerability to overcome resistance to CDK4/6 inhibitors in ER+/RB-deficient breast cancer.

## Methods
All experiments conducted in this study comply with relevant ethical regulations. Animal experiments were approved by the UTSW Institutional Animal Care and Use Committee (IACUC, protocol 2018-102359) and Department of the Army, Animal Care and Use Review Office (ACURO, protocol BC210406.e001). Mice were euthanized once the maximal tumor size (2,000 mm³, approved by IACUC and ACURO) was reached. De-identified tissue biopsy was collected from a patient under the study protocol BRE03103 (https://clinicaltrials.gov/study/NCT00899301) with IRB approval number 030747 at Vanderbilt University. The results of the trial have not been published yet. A written informed consent for the purpose of generating patient-derived xenografts (PDXs) was obtained.

### Cell lines and organoids
MCF-7 (Cat. No. HTB-22), T47D (Cat. No. HTB-133), HCC1428 (Cat. No. CRL-2327), ZR-75-1 (Cat. No. CRL-1500), and MDA-MB-436 (Cat. No. HTB-130) cells were purchased from ATCC. 293FT cells were purchased from Invitrogen (Cat. No. R70007). H596, H1048 and H1155 cell lines were kindly provided by Dr. John Minna. Du-145 cells were kindly provided by Dr. Ganesh Raj. CAMA1 and KPL1 cells were kindly provided by Dr. Benjamin Neel. Short tandem repeat (STR) profiling was used to verify authenticity of the cell lines. Cell lines were routinely tested for mycoplasma using MycoAlert mycoplasma detection kit (Lonza, Cat. No. LT07-710). MCF-7, MDA-MB-436 and 293FT cells were maintained in DMEM containing 10% FBS. T47D and Du-145 cells were maintained in RPMI containing 10% FBS. HCC1428, H596, H1048, H1155, and ZR-75-1 cells were maintained in RPMI containing 5% FBS. All culture media was supplemented with 1x antibiotic-antimycotic (Invitrogen). The PDxOs

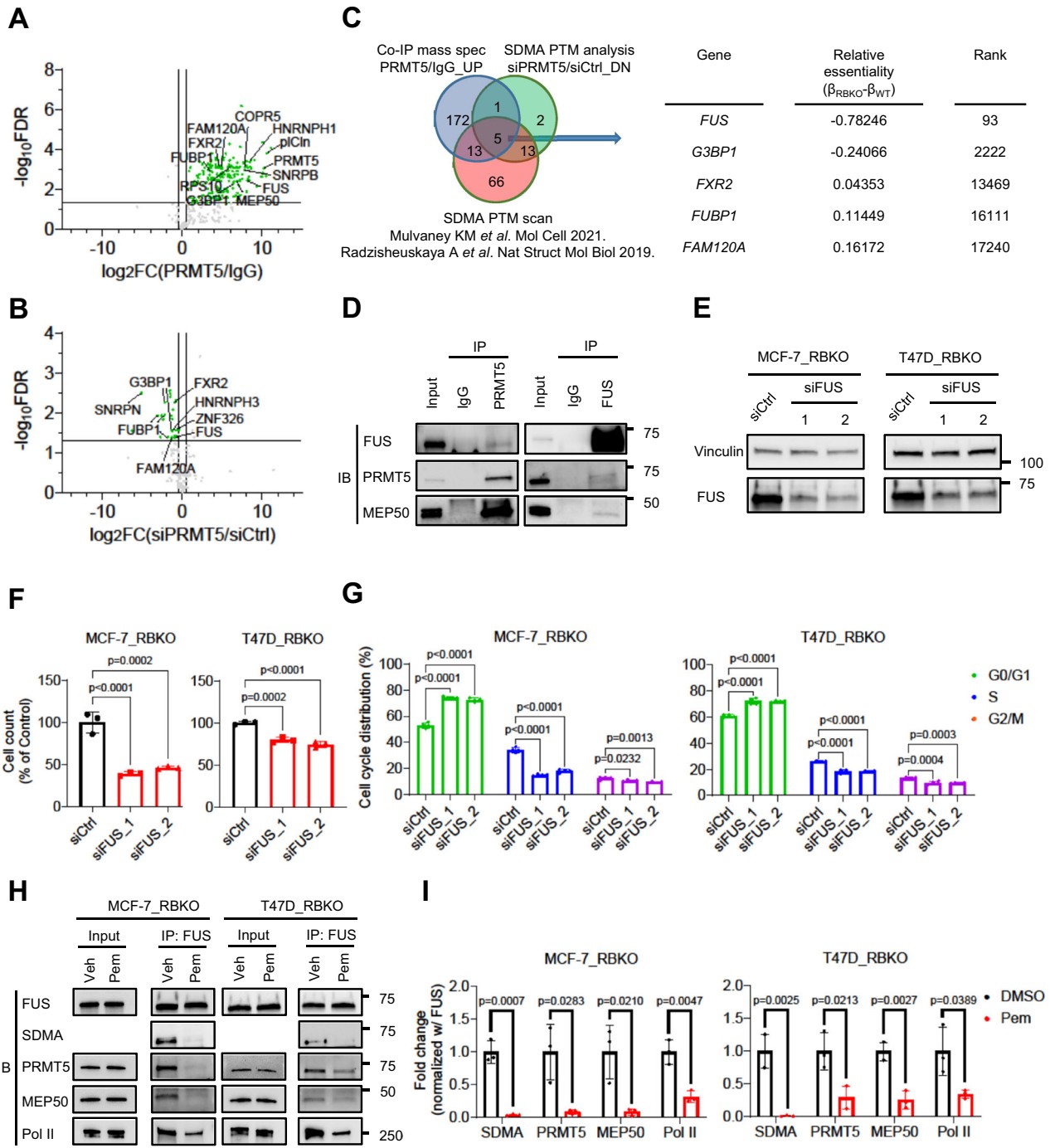

**Fig. 4 | Proteomics analysis identifies FUS as a downstream effector of PRMT5.**
**A** Co-immunoprecipitation (Co-IP) mass spectrometry (MS) analysis. PRMT5 antibody and rabbit IgG pulldowns of MCF-7_RBKO cell lysates were subjected to MS analysis. Green and grey points represent significant ($log_2FC \geq 0.5$ and FDR < 0.05) and non-significant ($log_2FC < 0.5$ or FDR > 0.05) enrichment of proteins, respectively. **B** SDMA post-translational modification (PTM) analysis. MCF-7_RBKO cells were transfected with *PRMT5* siRNA (siPRMT5) or control siRNA (siCtrl) for three days. Cell lysates were collected and subjected to IP using a SDMA antibody; antibody pulldowns were then subjected to LC-MS/MS analysis. Color codes are as described in (**A**). **C** Venn diagram integrating PRMT5 interacting proteins identified by Co-IP MS analysis, proteins where siPRMT5 reduced SDMA levels, and PRMT5 substrates identified by SDMA PTM analysis in published literatures. The table shows the common hits and their essentiality scores and ranking in the initial CRISPR screen. **D** Co-IP of MCF-7_RBKO lysates using a PRMT5 or a FUS antibody followed by immunoblot analysis (n = 2 biological replicates). **E** Immunoblot

analysis. Cell lysates were collected 3 days after transfection of siCtrl or two individual *FUS* siRNAs (siFUS) and then probed with antibodies as indicated (n = 2 biological replicates). **F** Monolayer growth assay. Cell number was counted using a Coulter counter five days after transfection of siCtrl (black) or siFUS (red). Data represent mean ± SD (n = 3 biological replicates), one-way ANOVA with a Dunnett's post-hoc test. **G** Cell cycle analysis. Cells were fixed 3 days after the transfection of siCtrl or two individual siFUS. The fixed cells were stained with propidium iodide and then subjected to flow cytometry analysis. Data represent mean ± SD (n = 3 biological replicates), one-way ANOVA with a Dunnett's post-hoc test. **H, I** Co-IP followed by immunoblot analysis. Cells were treated with DMSO or 200 nM Pemrametostat (Pem) for three days and then subjected to Co-IP using a FUS antibody. Immunoblot analysis of the FUS antibody pulldowns was conducted (**H**) and quantified (**I**) using the indicated antibodies. Data represent mean ± SD (n = 3 biological replicates), two-sided Student's t test. Source data are provided as a Source Data file.

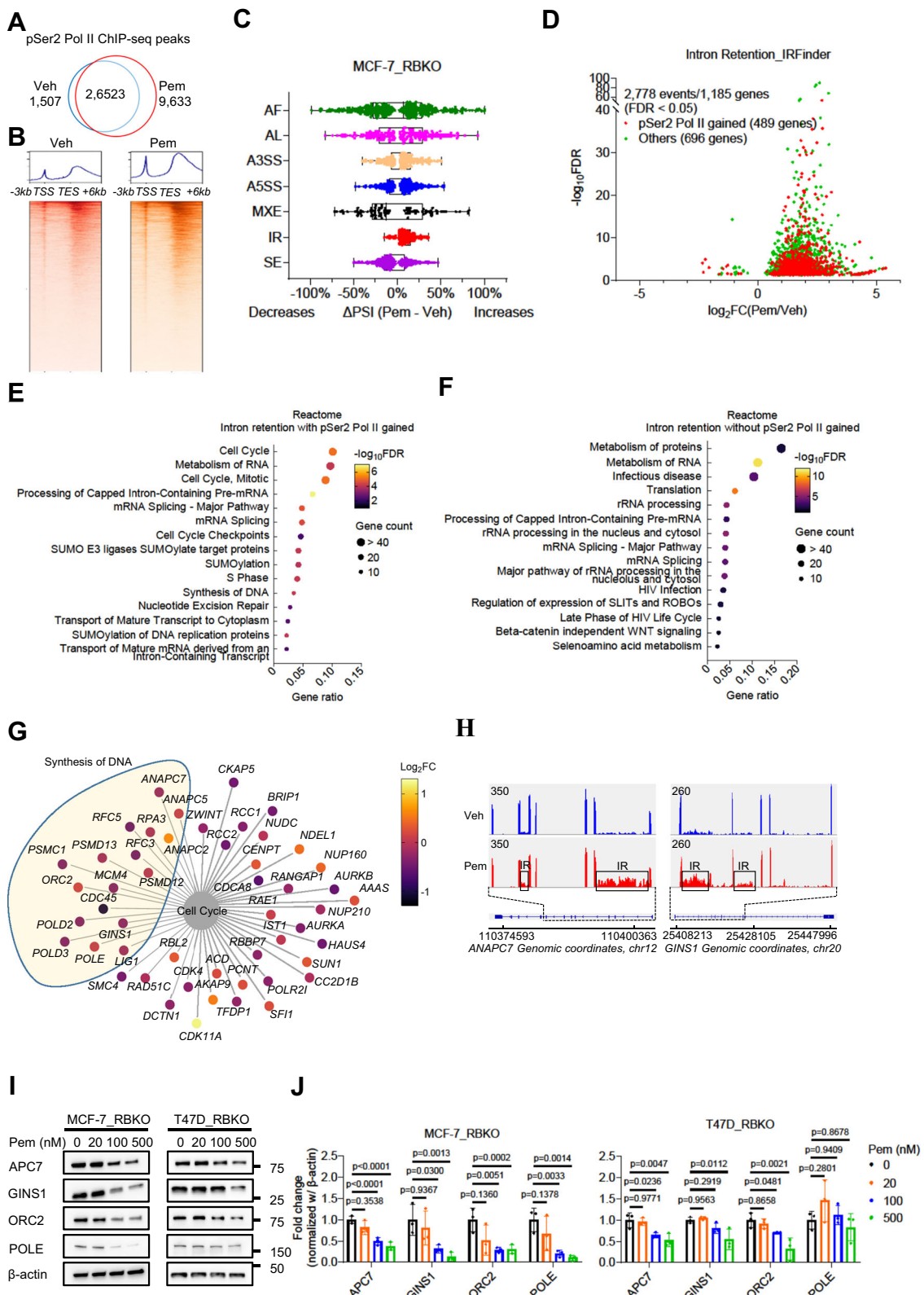

were maintained in Matrigel dome (Corning) supplemented with DMEM/F12 containing 250 ng/ml R-Spondin 3, 5 nM Heregulin β1, 5 ng/ml FGF7, 20 ng/ml FGF10, 5 ng/ml EGF, 100 ng/ml Noggin, 500 nM A83-01, 5 μM Y-27632, 500 nM SB202190, 1X B27 supplement, 1.25 mM N-Acetylcysteine, 5 mM Nicotinamide, 1X GlutaMax, 10 mM HEPES, 50 μg/ml primocin and 100 U/ml penicillin/ 100 μg/ml streptomycin.

**Plasmids**

pX458 and plentiCRISPR_v2 were gifts from Feng Zhang (Addgene plasmid #48138 and #52961). sgRNAs targeting *RB1* were subcloned into pX458 for the establishment of *RB1* knockout cells. Briefly, pX458 was digested with BbsI and ligated with oligonucleotides containing sgRNA sequences targeting *RB1*. For *PRMT5* depletion, sgRNAs targeting *PRMT5* were subcloned into plentiCRISPR_v2,

**Fig. 5 | PRMT5 inhibition results in Pol II Ser2 hyperphosphorylation and intron retention of cell cycle regulating genes. A** Venn diagram of pSer2 Pol II peaks identified by ChIP-seq in MCF-7_RBKO cells treated with 500 nM pemrametostat (Pem) or vehicle (Veh) for 72 hours (n = 2 biological replicates). **B** Heatmap displaying pSer2 Pol II binding intensity based on ChIP-seq in MCF-7_RBKO cells. *TSS*: transcription start sites; *TES*: transcription end sites. **C** Differences in percentage spliced in index (ΔPSI) between Pem- or Veh-treated MCF-7_RBKO cells. Analysis was conducted using SUPPA2 with RNA-seq data from MCF-7_RBKO cells treated with 500 nM Pem or Veh for 72 hours (n = 3 biological replicates). AF: alternative first exon; AL: alternative last exon; A3SS: alternative 3' splice-site; A5SS: alternative 5' splice-site; MXE, mutually exclusive exon; IR, intron retention; SE: skipped exon. Vertical lines within boxes represent median, edges of boxes represent the first or fourth quartiles, and whiskers represent the minimum or maximum values, outliers (greater or less than 1.5× interquartile range) were excluded. **D** IR events identified using IRFinder. The analysis was conducted with the same dataset as described in

(**C**). Transcripts with significant changes in IR were stratified based on whether their corresponding genes gained (red) or not (green) pSer2 Pol II chromatin bindings upon treatment of Pem. **E, F** Reactome pathway analysis using the gene stratification as described in (**D**). Pathway enrichment of genes that gained or not pSer2 Pol II chromatin bindings was shown in (**E**) and (**F**), respectively. **G** Visualization of genes enriched for cell cycle pathway as described in (**E**). Color code represents differential gene expression (Pem vs Veh). **H** Schematic of representative genes with IR induced by Pem. Upper left numbers denote exon/intron coverage on the same scale for each gene. *X* axis denotes genomic coordinates. **I, J** Immunoblot analysis of MCF-7_RBKO and T47D_RBKO lysates. Cells were treated with different concentrations of Pem for 4 days. Immunoblot analysis was conducted (**I**) and quantified (**J**) using the indicated antibodies. Data represent mean ± SD (n = 3 biological replicates), one-way ANOVA with a Dunnett's post-hoc test. Source data are provided as a Source Data file.

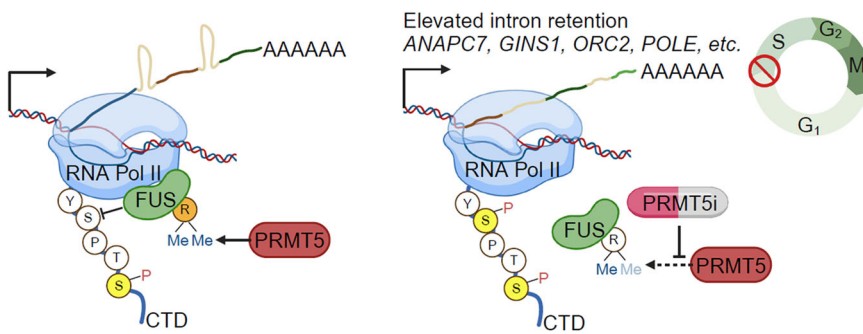

**Fig. 6 | Schematic of the proposed model.** PRMT5 inhibitors (PRMT5i) suppress symmetric dimethylation of arginine in FUS and uncouple FUS from RNA polymerase II (Pol II). The dissociation of FUS from RNA Pol II results

in hyperphosphorylation of Ser2 Pol II and intron retention of genes that promote DNA synthesis, thus blocking G1-to-S phase transition. Figure was created with BioRender.com.

which was digested with Esp3I and then ligated with oligonucleotides containing sgRNA sequences targeting *PRMT5*. Tet-pLKO-puro and Tet-pLKO-puro-Scrambled were gifts from Dmitri Wiederschain (Addgene plasmid #21915 and #110470). *p*LX304-zeo and *p*LX304-zeo-eGFP were gifts from Rizwan Haq (Addgene plasmid #160092 and #160095). shRNA targeting *PRMT5* 3'UTR was subcloned into Tet-pLKO-puro. Tet-pLKO-puro was digested with AgeI and EcoRI and then ligated with oligonucleotides containing shRNA sequence targeting *PRMT5* 3'UTR. *p*DONR221_PRMT5_WT was purchased from DNASU. Enzymatically dead *p*DONR221_PRMT5_E444Q was generated using the Q5 site-directed mutagenesis kit (NEB BioLabs). PRMT5_WT and PRMT5_E444Q open reading frames were subcloned into pLX304-zeo using LR Gateway clonase (Invitrogen). Virus packaging vectors *p*sPAX2 and *p*MD2.G were gifts from Didier Trono (Addgene plasmid #12260 and #12259). MISSION® pLKO.1-puro shRB1 plasmid (TRCN0000288710) and GFP shRNA control plasmid were purchased from Millipore-Sigma. Primers used for cloning are listed in Supplementary Table 1.

## Antibodies

For immunoblot analysis: Antibodies purchased from Cell Signaling include: PRMT5 Ab (2252 s, 1:1000), Total Rb (4H1) mAb (9309 s, 1:1000), pRB/s807/811 (D20B12) mAb (8516 s, 1:1000), β-actin (13E5) mAb (4970 s, 1:5000), Vinculin (E1E9V) mAb (13901 s, 1:2000), SDMA Ab (13222 s, 1:1000), MEP50 Ab (2823 s, 1:1000), PRMT5 (D5P2T) mAb (79998 s, 1:1000), and ORC2 (3G6) mAb (4736 S, 1:1000); Antibodies purchased from Santa Cruz Biotechnology include: ERα (F-10) mAb (sc-8002, 1:1000) and POLE (D-10) mAb (sc-390785, 1:1000); Antibodies purchased from Abcam: PRMT5 (EPR5772) mAb (ab109451, 1:1000) and GINS1 (EPR13359) mAb (ab181112, 1:1000); FUS Ab was purchased from Proteintech (11570-1-AP, 1:1000); Total Pol II (4H8) mAb was purchased from MilliporeSigma (05-623, 1:1000); APC7 Ab was purchased from Bethyl Laboratories (A302-551A, 1:1000). For Co-

IP: PRMT5 mAB (EPR5772) was purchased from Abcam (ab109451, 1:250); FUS mAB (4H11) was purchased from Santa Cruz Biotechnology (sc-47711, 1:100). For ChIP-seq: pSer2 Pol II Ab was purchased from Abcam (ab5095, 1:40). For IHC, SDMA Ab was purchased from Cell Signaling (13222 s, 1:600); ERα mAb (F-10) was purchased from Santa Cruz Biotechnology (sc-8002, 1:800); Ki67 (MIB-1) mAb was purchased from Agilent (IR62661-2, ready to use).

## *RB1* knockout

To mimic *RB1* loss-of-function alterations, we used CRISPR-Cas9 to knockout *RB1* in MCF-7 and T47D cells. This was achieved by transient transfection of *p*X458 plasmid carrying individual sgRNAs targeting *RB1* and followed by sorting for green fluorescence protein (GFP)-positive single cells using flow cytometry. RBKO single clones were validated by PCR-based genotyping, Sanger sequencing, and immunoblot analysis. GFP negativity was confirmed in the RBKO clones to ensure that the plasmid did not randomly integrate into the genome.

## siRNA transfection

Silencer Select siRNAs targeting *PRMT5* (ID: s20375 and s20377) and *FUS* (ID: s5401 and s533595) were purchased from Invitrogen. All Stars negative control siRNA was purchased from Qiagen. siRNA transfection was conducted using Lipofectamine RNAiMAX (Invitrogen).

## Lentiviral transduction

Virus packaging was conducted by co-transfection of *p*sPAX2 and *p*MD2.G plasmids along with viral vectors into 293FT cells. Media was replenished 24 hours after transfection, and virus supernatant was collected 24 hours later. Target cells were transduced with virus supernatant in the presence of 8 µg/mL polybrene and then selected with puromycin (1 µg/mL) or zeocin (100 µg/mL) according to the selection marker carried in the viral vectors.

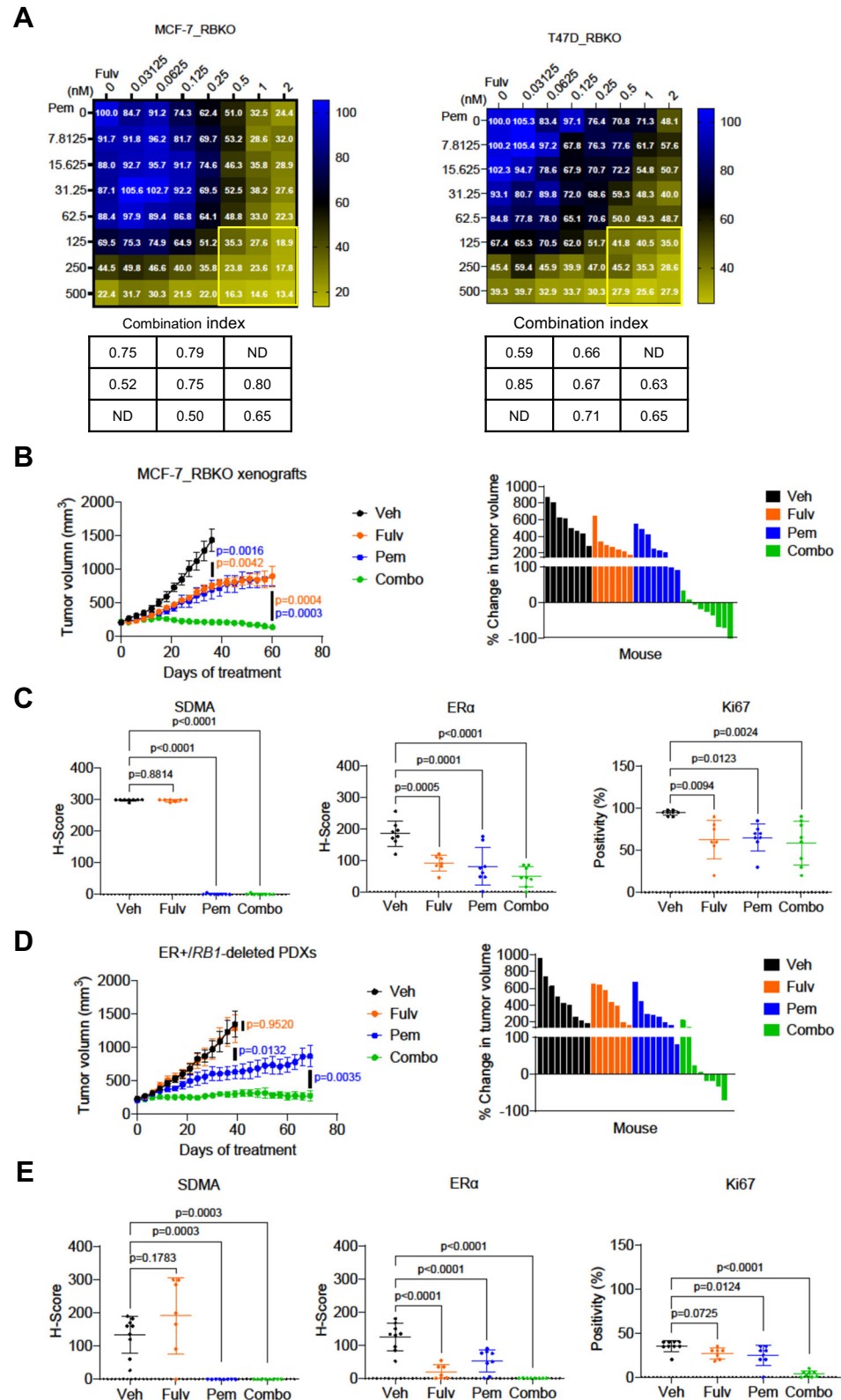

## Genome-wide CRISPR screen

Human Brunello CRISPR knockout pooled library was a gift from David Root and John Doench (Addgene #73178). The CRISPR screen was performed using T47D_WT and T47D_RBKO cells, each in two replicates. The lentiviral sgRNA library was transduced into T47D_WT and two independent RBKO clones at low multiplicity of infection (MOI = 0.3) and at a coverage of ≥500×. Non-transduced cells were eliminated by puromycin (1 μg/mL) selection for 7 days. After puromycin removal,

cells were collected on days 0 and 30. Genomic DNA from both time points was extracted. The sequences encoding the sgRNAs were PCR-amplified and then subjected to deep sequencing at the UTSW Next Generation Sequencing Core to determine sgRNA abundance.

## Cell viability assays

Cell viability of 2D-cultured cells was determined using a Z2 Coulter Counter Analyzer (Beckman) or by the CyQuant cell proliferation assay

**Fig. 7 | Combination of fulvestrant and pemrametostat synergistically inhibits growth of ER+/*RB1*-deficient breast cancer in vitro and in vivo. A** Monolayer growth of MCF-7_RBKO and T47D_RBKO cells treated with a dose range of fulvestrant (Fulv) and pemrametostat (Pem) for 6 days. Cell viability was determined by the CyQuant assay. The numbers in the heatmap represent averaged cell viability (% of control, n = 3 biological replicates). The table lists the combination indices of the highlighted region in the heatmap, with the index <1 representing synergistic effects. ND: not determined. **B)** Tumor volume of MCF-7_RBKO xenografts in female nude mice. The mice were treated with vehicle control (Veh, n = 8 mice), Fulv (5 mg/kg/week, s.c., n = 7 mice), Pem (200 mg/kg/day, p.o., n = 8 mice), or combination of both drugs (Combo, n = 9 mice). Data represent mean ± SD, one-way ANOVA with a Dunnett's post-hoc test; color codes represent statistical comparisons between individual treatment vs Veh or vs Combo. **C)** Quantification of immunohistochemistry (IHC) of the MCF-7_RBKO xenografts. Data represent mean ± SD (Veh n = 8 mice, Fulv n = 7 mice, Pem n = 8 mice, Combo n = 8 mice), one-way ANOVA with a Dunnett's post-hoc test. **D)** Tumor volume of ER + /*RB1*-deleted PDXs in female NSG mice. The mice were treated as described in (**C**). Data represent mean ± SD (Veh n = 9 mice, Fulv n = 7 mice, Pem n = 8 mice, Combo n = 8 mice), one-way ANOVA with a Dunnett's post-hoc test; color codes represent statistical comparisons between individual treatments vs Veh or vs Combo. **E)** Quantification of IHC of the PDXs. Data represent mean ± SD (Veh n = 9 mice, Fulv n = 7 mice, Pem n = 8 mice, Combo n = 8 mice), one-way ANOVA with a Dunnett's post-hoc test. Source data are provided as a Source Data file.

(Invitrogen). Cell viability of 3D-cultured organoids was determined by the 3D Cell TiterGlo assay (Promega) on a GloMax plate reader (Promega).

## Cell cycle analysis
Cells were fixed with ice-cold 70% ethanol and stored at −20 ℃. On the day of cell cycle analysis, fixed cells were washed twice with ice-cold PBS and then stained with 50 μg/mL propidium iodide (Invitrogen) in PBS supplemented with 10 μg/mL RNase A (Invitrogen). Cell cycle analysis was performed using a LSRFortessa flow cytometer (BD Biosciences).

## RNA-seq
Total RNA was extracted using Maxwell RSC simply RNA kit (Promega). mRNA libraries were prepared using TruSeq Stranded mRNA Library prep kit (Illumina) and sequenced on NextSeq 550 sequencer (Illumina) in a PE75 run. RNA-Seq analysis was conducted by aligning sequencing reads to the human reference genome GRCh38 using STAR[58], and gene expression levels were estimated as row read counts. DESeq2 was used to assess the statistical significance of differentially expressed genes[59]. Gene set enrichment analysis (GSEA) and gene ontology (GO) analysis were conducted using the GSEA software[60] and DAVID[61], respectively. RNA splicing analysis was first conducted using SUPPA2[62], and further analysis to identify IR was conducted using IRFinder[43].

## qRT-PCR
Total RNA was reversely transcribed to cDNA using iScript kit (Bio-Rad). qRT-PCR was performed using SYBR Green master mix (ThermoFisher) on a QuantStudio 3 Real-Time PCR System (ThermoFisher). Expression of *YWHAZ* was used as internal control for normalization. Sequences of primers are listed in Supplementary Table 1.

## Immunoblot analysis and co-immunoprecipitation
For immunoblot analysis, cells were lysed with RIPA buffer supplemented with proteinase and phosphatase inhibitors (Roche). Protein concentration was determined using Gold Rapid BCA (Thermo Fisher). Proteins were separated by 4-20% gradient SDS-PAGE (Bio-Rad) or 4-12% NuPage gradient gels (Invitrogen), transferred onto nitrocellulose membrane, blocked with 5% non-fat milk, and then probed with primary antibodies. HRP-conjugated anti-rabbit or anti-mouse were used as secondary antibodies. For Co-IP experiments, cells were lysed with NP-40 lysis buffer (20 mM Tris-HCl pH 7.6, 150 mM NaCl, 0.1% NP-40, 1 mM EDTA) supplemented with proteinase and phosphatase inhibitors. Protein concentration was determined using the Gold Rapid BCA. Lysates were pre-cleared with protein-G-conjugated Dyna beads (Invitrogen) and then incubated with either a PRMT5 or a FUS antibody or a control IgG overnight at 4℃. Next day, lysates were incubated with protein-G-conjugated Dyna beads for 2 hours at 4℃, washed three times with NP-40 lysis buffer, and then eluted with 0.2 M Glycine (pH 2.0). Eluates were subjected to SDS-PAGE and immunoblot analysis. For Co-IP MS

analysis, immunoprecipitated proteins were further processed by running in a SDS-PAGE gel followed by Coomassie blue staining, tryptic digestion, reduction with DTT, alkylation with iodoacetamide, and finally cleanup with Oasis MCX solid-phase extraction cartridges (Waters).

## Immunoaffinity enrichment of peptides containing SDMA
Cells were lysed with PTMScan® Urea Lysis Buffer (9 M urea, 20 mM HEPES pH 8.0; Cell Signaling) and then sonicated at 25% amplitude using a microtip sonifier (Branson 150). Protein concentration was determined using Gold Rapid BCA, and lysates were reduced and alkylated with DTT and iodoacetamide, respectively. Next, lysates were subjected to tryptic digestion, solid-phase cleanup and SDMA enrichment following the manual instruction of PTMScan® Symmetric Di-Methyl Arginine Motif [sdme-RG] Kit (Cell Signaling). SDMA enriched samples were subjected to secondary digestion, solid-phase cleanup, and finally analyzed by LC-MS/MS.

## MS and data analysis
MS data were acquired using a Q-Exactive HF Quadrupole-Orbitrap mass spectrometer (Thermo Fisher) for the PRMT5 antibody pulldowns and an Orbitrap Fusion Lumos Tribrid mass spectrometer (Thermo Fisher) for the SDMA enriched samples. Data were analyzed using Proteome Discoverer 2.4 and were searched using the human protein database from UniProt. Proteins were filtered for downstream analysis using a cutoff FDR < 0.01 with at least two peptides being mapped.

## ChIP and ChIP-seq
Cells were fixed with 1% formaldehyde for 10 min and then quenched with 125 mM glycine for 5 min at room temperature. Cells were next washed twice with ice-cold PBS and harvested by scrapping. Cells were lysed with PIPES buffer (5 mM PIPES pH 8.0, 85 mM KCl, 0.5% NP-40; Santa Cruz Biotechnology), and nuclei were isolated by centrifugation and then lysed with ChIP high salt lysis buffer (PBS, 1% NP-40, 0.5% Sodium Deoxycholate, 0.1% SDS; Santa Cruz Biotechnology). Both PIPES and ChIP high salt lysis buffer were supplemented with proteinase and phosphatase inhibitors. Chromatin was sheared using a microtip sonicator (Branson 150) to an average fragment size of 100-300 bp. Sheared chromatin containing 25 μg DNA was diluted with IP buffer (0.01% SDS, 1.1% Triton-X, 1.2 mM EDTA, 16.7 mM Tris-HCL pH 8.0, 167 mM NaCl) and incubated overnight at 4 °C with 10 μg of precipitating antibodies. The next day, antibody pulldowns were incubated with protein-G-conjugated Dyna beads for 2 hours at 4℃. The beads were washed once with each of the following buffers in sequence: low salt (0.1% SDS, 1% Triton X-100, 2 mM EDTA, 20 mM Tris-HCl pH 8.0, 150 mM NaCl), high salt (0.1% SDS, 1% Triton X-100, 2 mM EDTA, 20 mM Tris-HCl pH 8.0, 500 mM NaCl), LiCl wash (0.25 M LiCl, 1% NP40, 1% deoxycholate, 1 mM EDTA, 10 mM Tris-HCl pH 8.0), and Tris-EDTA (pH 8.0). DNA was eluted with elution buffer (0.1 M NaHCO3, 1% SDS) for 1 hour at 65℃ and then incubated with 200 mM NaCl overnight at 65℃ to reverse crosslinking. Next day, the elution was treated with RNase A (Thermo Fisher) and Proteinase K

(Thermo Fisher); DNA was then purified using ProNex Size-Selective Purification System (Promega). For ChIP-seq, libraries were prepared using the Kapa HyperPlus Kit (Roche) and sequenced using Illumina NextSeq 550 sequencer with PE-75.

ChIP-Seq analysis was conducted by aligning sequencing reads to the human reference genome GRCh37 using Bowtie2[63] with default parameters. Peaks were called using Model-based Analysis of ChIP-Seq (MACS) software[64] with default parameters and FDR < 0.05 as cutoff.

## Xenograft studies

Six weeks old female athymic nude-*Foxn1*[nu] mice and NOD-*scid* IL2Rgamma[null] (NSG) mice were purchased from Envigo. All mice housed in barrier facilities were maintained in individually ventilated microisolator cages. All caging equipment was autoclaved and all feed was commercial irradiated diet. Cage manipulations and animal handling was performed in cage change stations or biosafety cabinets. Automated watering systems provided water that was purified through reverse osmosis and chlorination. The standard white light cycle was from 6:00 AM to 5:59PM and the dark cycle was from 6:00PM to 5:59AM. An estrogen pellet (0.25 mg/pellet, 21-day release; Innovative Research of America) was implanted s.c. in the mouse dorsum one day before tumor inoculation. One million MCF-7_RBKO cells were mixed in PBS:matrigel (1:1) and then injected s.c. into 6-week-old female nude mice. For the PDX model, the de-identified tissue biopsy from a patient with ER+ metastatic breast cancer acquired resistance to letrozole plus palbociclib was collected. Deletion of *RB1* in the biopsy was reported by Foundation One biomarker testing. The biopsy was implanted s.c. into female NSG mice to establish the PDX model. Genomic DNA extracted from the PDX was subjected to whole-exome sequencing at UTSW Sequencing Core. For copy number analysis, we randomly selected 10 whole-exome sequenced samples from normal breast tissue in TCGA as a synthetic reference for germline alterations. Deletion of *RB1* in the PDX was confirmed by comparing *RB1* copy number of the PDX to that of the synthetic reference. The ER + /*RB1*-deleted PDX fragments were implanted s.c. into 6-week-old female NSG mice one day after implantation of estrogen pellets. Once tumors reached ≈200 mm$^3$, mice were randomized to receive treatment with vehicle (5% DMSO, 40% PEG-300 and 5% Tween-80 in sterilized water), fulvestrant (5 mg/mouse/week, s.c.), pemrametostat (200 mg/kg/day, via orogastric gavage), or combination of both drugs. Tumor size was serially measured with calipers and calculated every three days with the formula: volume = width$^2$ × length/2. At the end of the treatment, tumors were harvested and then snap frozen in liquid N$_2$ or fixed in 10% neutral-buffered formalin.

## IHC

Formalin-fixed tumors were embedded in paraffin; 5-μm tumor sections were used for IHC. Nuclear positivity of Ki67 and H-scores of SDMA and ERα were quantified by an expert breast pathologist blinded to treatment arms and using standard CAP breast tumor biomarkers scoring guidelines. For cells from the PDxOs, cells in a Matrigel dome were fixed in 10% neutral-buffered formalin for 1 hour at room temperature, pelleted in 2.5% low-melt agarose, and then subjected to paraffin embedding, sectioning, and IHC.

## Statistics & reproducibility

Statistical analysis was performed using GraphPad Prism 9. The comparison between two groups was analyzed by two-sided Student's t test, while multi-group comparisons were analyzed by one-way ANOVA with a Dunnett's post-hoc test, both methods with 95% confidence intervals and (n-1) degree of freedom. All experiments were conducted at least two independent times with similar results if not otherwise specified. In general, sample size was determined based on standards for cell line and animal studies from our previous published studies (PMID: 33127913). No data were excluded from the analyses. For all

experiments, subjects were randomly assigned to experimental groups. For IHC quantification analysis, a breast cancer pathologist was blinded to treatment groups and performed quantification for the staining. Blinding was not possible for other experiments as the investigators must be aware of what treatment to give cells and animals.

## Reporting summary

Further information on research design is available in the Nature Portfolio Reporting Summary linked to this article.

## Data availability

All data associated with this study are presented in the paper or Supplementary Information. Source data are provided with this paper. Raw RNA-seq and ChIP-seq data generated in this study have been deposited in the Gene Expression Omnibus under accession code GSE236500. Raw MS data generated in this study were deposited in the ProteomeXchange Consortium via the PRIDE partner repository under accession code PXD046996. Human reference genome assembly GRCh37 and GRCh38 were used for data analysis. Source data are provided with this paper.

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

## Acknowledgements
We acknowledge the assistance of the Tissue Management Shared Resource and the Proteomics Core of the Simmons Comprehensive Cancer Center. This work is supported by, National Cancer Institute P30 CA142543 (C.L. Arteaga), DOD BC 210406 (C.C. Lin), NCI R01CA224899 (C.L. Arteaga and A.B. Hanker), NCI Breast SPORE P50 CA098131 (C.L. Arteaga and A.B. Hanker), CPRIT RR170061 (C.L. Arteaga), Susan G. Komen Breast Cancer Foundation SAB1800010 (C.L. Arteaga), Breast Cancer Research Foundation DRC-20-001 (C.L. Arteaga and A.B. Hanker), CPRIT RP220309 (J.T. Mendell), and The Welch Foundation I-1961-20210327 (J.T Mendell). J.T Mendell is an Investigator of the Howard Hughes Medical Institute.

## Author contributions
Experimental design and conception: C.C. Lin, T.S. Chang, E. Bikorimana, A. Lemoff, Y.V. Fang, D. Ye, I. Soria-Bretones, A. Servetto, K.M. Lee, X. Luo, J.J. Otto, H. Akamatsu, F. Napolitano, R. Mani, Cescon DW, L. Xu, Y. Xie, J.T. Mendell, A.B. Hanker, and C.L. Arteaga. Data analysis: C.C. Lin, Wang Y, L. Guo, Y. Gao, A. Lemoff, Y.V. Fang, H. Zhang, Y. Zhang, and K.M. Lee. Writing and revision of the manuscript: C.C. Lin and C.L. Arteaga.

## Competing interests
J.T.M. is a scientific advisor for Ribometrix, Inc., and owns equity in Orbital Therapeutics, Inc. H.A. receives honoraria from Boehringer Ingelheim Japan Inc., Eli Lilly Japan K.K., and Taiho Pharmaceuticals. A.B.H. received or has received research grants from Takeda and Lilly and nonfinancial support from Puma Biotechnology and Daiichi Sankyo. C.L.A. receives or has received research grants from Pfizer, Lilly, and Takeda; holds minor stock options in Provista; serves or has served in an advisory role to Novartis, Merck, Lilly, Daiichi Sankyo, Taiho Oncology, OrigiMed, Puma Biotechnology, Immunomedics, AstraZeneca, Arvinas, and Sanofi; and reports scientific advisory board remuneration from the Susan G. Komen Foundation. D.W.C. reports advisory services to AstraZeneca, Exact Sciences, Eisai, Gilead, GlaxoSmithKline, Inivata, Merck, Novartis, Pfizer, and Roche; reports research funding (to institution) from AstraZeneca, Gilead, GlaxoSmithKline, Inivata, Merck, Pfizer, and Roche; and is a member of a trial steering committee for AstraZeneca, Merck, and GlaxoSmithKline. Other authors declare no competing interests.

## Additional information

[1]Harold C. Simmons Comprehensive Cancer Center, UT Southwestern Medical Center, Dallas, TX, USA. [2]Department of Molecular Biology, UT Southwestern Medical Center, Dallas, TX, USA. [3]Howard Hughes Medical Institute, UT Southwestern Medical Center, Dallas, TX, USA. [4]Quantitative Biomedical Research Center, Department of Population & Data Sciences, Peter O'Donnell Jr. School of Public Health, UT Southwestern Medical Center, Dallas, TX, USA. [5]Division of Pediatric Gastroenterology, Hepatology and Nutrition, Cincinnati Children's Hospital Medical Center, Cincinnati, OH, USA. [6]Department of Pathology, UT Southwestern Medical Center, Dallas, TX, USA. [7]Department of Biochemistry, UT Southwestern Medical Center, Dallas, TX, USA. [8]Department of Genetics, University of Alabama at Birmingham, Birmingham, AL, USA. [9]Princess Margaret Cancer Centre, University of Toronto, Toronto, ON, Canada. [10]Department of Clinical Medicine and Surgery, University of Naples Federico II, Naples, Italy. [11]Department of Life Science, Hanyang University, Seoul, South Korea. [12]Third Department of Internal Medicine, Wakayama Medical University, Wakayama, Japan. ✉e-mail: Carlos.Arteaga@UTSouthwestern.edu

