## [Peer Review File · Nature Communications]

Reviewers' Comments:

Reviewer #1:

Remarks to the Author:

The manuscript submitted by Lin et al. demonstrates the viability of PRMT5 as a target for breast cancer that is ER+ but resistant to CDK4/6 inhibitors. The study confers CDK4/6i resistance by a knockout of RB1, RBKO, and an sgRNA screen revealed hits for new vulnerabilities despite the drug resistance. Choosing PRMT5, the study finds knockout or small molecule inhibitors can stop growth and block cell cycle for RBKO breast cancer cell lines. Lastly, inhibition of PRMT5 and block of ER activity synergistically to slow or reduce tumor volume.

The study is thorough, and all results generally point in the same direction that RBKO cells are susceptible to inhibition of PRMT5. The study findings show PRMT5i activity in RBKO breast cancer cells that matches well with observations in other cancer types. What appears novel or unexpected in this manuscript is the response of breast cancer to PRMT5i after being resistant to CDK4/6i, the effect of PRMT5i on pSer2, and the synergy between PRMT5i and fulvestrant. These experiments are well done, described thoroughly, and should be published. The supplemental Excel files are nicely done and easy to follow. However, there are corrections or areas that need to be addressed.

1. Some labels need to be corrected. Figure 2C is mislabeled, the left 2 lanes were presumably not treated with shPRMT5. The labels for Enrichment Plots in Figure 3A and Supp Figure 3 are almost illegible and should be replaced with text. The order is swapped for the enrichment plots shown for MCF7_RBKO and T47D_RBKO in Figure 3A, which is confusing.

2. The complex figures require lengthy legends that can be difficult to navigate. Additional labeling within the figure would help the reader follow what is shown in the figures: the drug name in Figure 2F, molecular weight markers in Figure 4H and/or full gel image in the supplement showing where the SDMA band runs, label for what data is plotted in Figures 5E and F.

3. There's considerable redundancy in the ontology and association analysis data shown. For example, in Figure 5E the proteins in "SUMO E3 ligases..." seems likely to be most or all the same proteins as "SUMOylation" on the line below. There are further multiples of "Cell cycle..." and "mRNA" in Figure 5E and 5F. The data in the main figures is also crowded with noise and irrelevance, such as "HIV infection" proteins for MCF7 cells in Figure 5F, or "Spermatogenesis" and "Heme_metabolism" in MCF7 and T47D cells in Figure 3A. The full lists of associations would be nice for the supplement tables, but they make it difficult to find what the authors describe in their results and the text is near impossible to read. It might also convey incorrect meaning, for example the three "methyltransferase" associations in red for Figure 1C highlight an inherent flaw in ontology analysis but does not make the association 3 times more important.

4. It's difficult to see the change described in the text and caused by siPRMT from the stacked plot in Figure 3D. For this instance of the cell cycle data, a bar plot and significance indicated. Other stacked bar plots in the manuscript seem fine.

5. The bottom line about FUS seems similar to the earlier reports the authors cite demonstrating PRMTi effects on SR and hnRNP protein activities (PMID: 31611688, 33782401, 29946150). In fact, previous studies find evidence that FUS, hnRNP proteins, and SR proteins bind each other and are recruited together to the initiating and elongating RNA polymerase (PMID: 9774382, 22855529, 26124092). However, Figure 5J implies more than this or other studies have shown. The authors imply intron inclusions, like those shown in 5H, could explain reduced protein, Figure 5I. How many reduced proteins and increased introns correlate is measurable by whole cell proteomics, showing whether Figure 5I changes are the only options to affect cell cycle or among many scores of proteins similarly diminished. An example of one alternative model is that widespread stalling of RNA Pol II can interfere with both splicing and cell cycle, but in parallel. The same cell cycle genes of interest aren't known to be affected the same way by PRMT5i and loss of FUS. The SDMA PTM scan uncovered more RNA processing factors closely related to FUS. More scrutiny of FUS activity does not improve this study. The data is strong that targeting FUS activity validates and supports specificity of PRMT5i and FUS activity must overlap a significant portion of

the mechanism of PRMT5i. It raises excitement for more study in the future of FUS and whether its activity on cell cycle can be linked to a collection of introns affected.

6. The inset table in Figure 6A is unclear. Why there is different shading is not explained. These have labels in the figure. It would be interesting if the authors had tested ERE luciferase activity in the presence of PRMT5i.

Reviewer #2:

Remarks to the Author:

In this paper the authors conclude that there is a potential role of PRMT5 inhibition to treat ER+ breast cancer patients who have failed to respond to CDK4/6 inhibitors due to loss of the RB protein. This represents a small proportion of metastatic ER+ breast cancers, although they are undoubtedly difficult to treat. The molecular findings regarding PRMT5 function are interesting. However there are multiple weaknesses in this paper described in the points below, many of which limit the observations to the models presented, rather than demonstrating real applicability in the disease setting that the authors are proposing.

1. Application to CDK4/6 inhibitor resistant ER+ breast cancer and selection of models

The authors conclude that "Collectively, these data support PRMT5 as a therapeutically actionable vulnerability to overcome resistance to CDK4/6 inhibitors in ER+/RB-deficient breast cancer." The authors use two main models, MCF-7 and T-47D cells, and a smattering of other models. Figures 4 and 5 exclusively describe MCF-7 cells. The models do not represent genuine CDK4/6 inhibitor resistance models that have arisen due to exposure to CDK4/6 inhibitors, but instead knockout RB models. The authors have used only one model with acquired CDK4/6 resistance (the PDX model in Figure 6), and that model is poorly described (see below). While the conclusion that RB KO ER+ cells are susceptible to PRMT5 inhibitors may be valid, the conclusion that CDK4/6 inhibitor resistant ER+ breast cancer with RB deficiency can be treated with PRMT5 inhibitors has not been shown.

This is an important distinction as there is no evidence that PRMT5 expression is maintained in RB-deficient ER+ breast cancer with resistance to CDK4/6 inhibitors. The authors could address this problem by (i) supplementing with genuine resistance models (in vitro and PDX) and (ii) examining the expression of PRMT5 in RB deleted ER+ metastatic breast cancers that are resistant to CDK4/6 inhibitors, preferably a patient cohort.

Particular issues with models are:

(A) The PDX model used in Figure 6D is not described adequately.

The current description is "RB1-deleted patient-derived xenografts (PDX) derived from a patient with ER+ MBC that progressed clinically on palbociclib plus the aromatase inhibitor letrozole. RB1 deletion in the PDX was confirmed by exome sequencing (data not shown)".

- Where does this PDX come from? What human ethics approvals support its development and use?
- How long was the patient treated for? Does this represent intrinsic or acquired resistance?
- Is this PDX model actually resistant to palbociclib? The implication is that this model represents palbociclib resistance, but it is not shown in the data. PDX models often behave differently to how the disease responds in the patient.
- Was the RB1 deletion in the patient or only in the PDX?
- Exome sequencing of the RB1 deletion by "data not shown" is inadequate.

(B) Cell line model selection.

The manuscript alternates between different models, and many of the findings are in one cell line only. For example the rescue experiments are T47D cells only (Figure 2D), and the mass spec/ChIP-seq is performed exclusively in MCF7 (Figure 4/5). The FUS and Pol II data needs validation in T-47D RBKO or another model (or even better, show high FUS/activity in samples from RB KO resistant patients), as otherwise this third of the paper (Fig4/5) relies upon a single cell line. The authors generalize the FUS/Pol II data to all ER+ breast cancer, but this isn't currently supported.

2. RB deletion versus RB deficiency

The authors have conflated the concepts of RB deficient and RB deleted in their paper. RB deletion

and truncating/inactivating mutations are relatively rare in metastatic ER+ breast cancer, occurring in 3-4% of cases. The current evidence suggests that they are not acquired at high frequency in patients treated with combination endocrine therapy and CDK4/6 inhibitors. See for example "RB1 mutations arise following treatment with CDK4/6 inhibition, but .. these mutations are likely subclonal and of relatively low prevalence, suggesting, in contrast to previous work, that they are not a major mechanism of resistance." doi: 10.1158/2159-8290.CD-18-0264 Heterozygous loss (<https://doi.org/10.1038/s41467-022-32828-6>) forms part of the RB deficient phenotype and many patients may have low RB or low RB function rather than complete loss. The authors' work exclusively models complete ablation of RB function by genomic deletion. As such they are describing a genotype affecting a small proportion of patients. However their language implies a much larger group of patients by using the description "RB1 deficiency". The evidence presented in the paper does not support that PRMT5 inhibition is useful in a context of low RB function, but only that it may be useful in the context of complete RB knockout. To support these conclusions regarding RB deficiency the authors need to show that their work applies in conditions of low RB function.

3. Some CDK4/6i resistant cancers present with downregulation of ER or a basal-like phenotype which is not hormone sensitive (and RB loss can occur in this setting). The models presented by the authors are universally ER+ and responsive to endocrine therapies with the exception of the PDX model. PRMT5 is suggested to act as an oncogene in endometrioid adenocarcinoma via the action of estrogen receptor <https://doi.org/10.1002/cjp2.194>. Are PRMT5 inhibitors going to be effective in CDK4/6i resistant cancers that have lost their reliance on ER?
4. Line 98-100 " CCND1 ($\beta_{WT} = -1.93$; $\beta_{RBKO} = -0.23$) and CDK4 ($\beta_{WT} = -1.75$; $\beta_{RBKO} = -0.67$) were not essential in RBKO compared to WT cells, consistent with the notion that loss of RB1 uncouples the CDK4/Cyclin D1 complex from E2F-regulated transcription and the G1-to-S transition" What is the definition of essentiality? This needs to be defined and justified.
5. Figure 2B is confusing for the sgPRMT5 MCF7 and T47D cells. How were these cell lines derived and plated for the experiment if they are unable to proliferate, as shown in Figure 2B?
6. Figure 2F/Supp Fig 2 – what is the sensitivity of RB expressing cells to the PRMT5 inhibitor? Since the CRISPR screen shows that RB wildtype cells are less reliant on PRMT5, a reduced sensitivity should be apparent with dose response assays.
7. All cell cycle data needs statistical analysis. Why don't all the cell cycle distribution experiments add up to 100%? Where is the gating strategy data?
8. Fig 4 – The authors assert that FUS knockdown phenocopies PRMT5 knockdown. This would be more convincing with some sort of rescue experiment. For example, does FUS knockdown cause cell cycle effects in cells with re-expressed WT PRMT5, but not in PRMT5 E444Q cells (as per Figure 2D)?
9. HCI-018 is used in this paper, but no evidence is presented that it is RB-deleted. This is needed. Also, is this model resistant to palbociclib?
10. Figure 5I: Are these proteins downregulated due to pemrametostat treatment, or due to off-target effects? Downregulation is only convincing at the high dose in Fig 5I (500nM, which is 3x the IC50), but not at the low dose, which raises the likelihood of off-target effects. Are these same proteins downregulated with PRMT5 shRNA or siRNA? I would find this whole figure more convincing if intron retention was also shown with PRMT5 knockdown.
11. Many other experiments are carried out with 500nM pemrametostat, and these doses also seem high, given that this dose leads to 25% cell viability after 6 days treatment. What is the overlap between the RNAseq on MCF7 RBKO treated with pemrametostat performed in Fig 5C, and PRMT5 siRNA RNAseq on MCF7 RBKO in Figure 3? If pemrametostat is being applied at a dose that mainly acts through PRMT5 then there should be excellent overlap between these datasets.
12. No uncropped western blots are shown. Densitometry of replicate western blots experiments

are lacking eg 2C, 2F, 5I.

Reviewer #3:

Remarks to the Author:

The paper by Lin et al describes the discovery of the protein arginine methyltransferase PRMT5 as a potential therapeutic target in CDK4/6 inhibitor-resistant breast cancer that are ER+/RB-deficient. This well-structure and composite paper addresses an important clinical issue, provides convincing mechanistic insights on PRMT5 activity in the tumor subtype under investigation and is very well written. The findings are largely compelling and appropriate for the wide audience of the journal. Therefore I recommend publication of the manuscript after minor revisions.

Few minor concerns should be considered prior to publication:

1) The authors performed a genome-wide CRISPR screen on both WT and RBKO cells and defined genes essential for one or both conditions. However, PRMT5 is essential for both. While there is no doubt that PRMT5 is a valuable target in RBKO, the authors should better discuss its potential role in WT cells.

1) In Figure 1D PRMT1 is also highlighted, but is not cited in the text: since PRMT1 is the most active type-I PRMT, with wide range of non-histonic protein targets and experimentally-reported cross-talk with PRMT5, it would be important to discuss this piece of evidence in the context of the story presented.

2) more information regarding the number of technical/biological replicates (for instance for western blots experiments) should be provided in the figure legends.

3) the statistical significance should be added in Figs 2B, 3D, 3E, 4G, 6B, 6D, S4C, S5C.

4) the x axis legend should be added in Fig. 6B-D (right plot)

5) the list of co-enriched proteins in panel 4A and of R-methylated peptides down-regulated upon siPRMT5 in figure 4B should be reported (besides uploading the MS raw data in open access repository), to gain a more complete view of the quantitative proteomic data acquired. The volcano-plot displayed in panel 4B (with absolutely not a single di-methylated peptide that are upregulated upon siPRMT5) is a bit surprising, in light of the well-documented scavenging elicited through PRMT1 when PRMT5 is inactivated). all in all, the proteomics data are presented in a too simplistic manner to be fully convincing: more data (in terms of txts data output derived by the quantitative analysis of MS raw data and statistical analysis applied) should be provided.

Dear Reviewers,

We sincerely appreciate the insightful and helpful critique of our study. We have meticulously reviewed your comments and revised our manuscript accordingly. Please see below for our point-by-point response to each of your comments, indicating the changes we have implemented. Thank you again for your time, valuable insights, and consideration.

REVIEWER COMMENTS

Reviewer #1 (Remarks to the Author):

The manuscript submitted by Lin et al. demonstrates the viability of PRMT5 as a target for breast cancer that is ER+ but resistant to CDK4/6 inhibitors. The study confers CDK4/6i resistance by a knockout of RB1, RBKO, and an sgRNA screen revealed hits for new vulnerabilities despite the drug resistance. Choosing PRMT5, the study finds knockout or small molecule inhibitors can stop growth and block cell cycle for RBKO breast cancer cell lines. Lastly, inhibition of PRMT5 and block of ER activity synergistically to slow or reduce tumor volume.

The study is thorough, and all results generally point in the same direction that RBKO cells are susceptible to inhibition of PRMT5. The study findings show PRMT5i activity in RBKO breast cancer cells that matches well with observations in other cancer types. What appears novel or unexpected in this manuscript is the response of breast cancer to PRMT5i after being resistant to CDK4/6i, the effect of PRMT5i on pSer2, and the synergy between PRMT5i and fulvestrant. These experiments are well done, described thoroughly, and should be published. The supplemental Excel files are nicely done and easy to follow. However, there are corrections or areas that need to be addressed.

1. Some labels need to be corrected. Figure 2C is mislabeled, the left 2 lanes were presumably not treated with shPRMT5. The labels for Enrichment Plots in Figure 3A and Supp Figure 3 are almost illegible and should be replaced with text. The order is swapped for the enrichment plots shown for MCF7_RBKO and T47D_RBKO in Figure 3A, which is confusing.

We apologize for mislabeling Figure 2C and the small font size in the GSEA plots. We have revised the figures according to your suggestions.

Fig. 2C

Fig. 3A

Supplementary Fig. 4A

Supplementary Fig. 4B

2. The complex figures require lengthy legends that can be difficult to navigate. Additional labeling within the figure would help the reader follow what is shown in the figures: the drug name in Figure 2F, molecular weight markers in Figure 4H and/or full gel image in the supplement showing where the SDMA band runs, label for what data is plotted in Figures 5E and F.

Thank you for the comment. We have revised Figures 2F, 5E and F according to your suggestions. For Figure 2F, we added drug name and included data of WT cells. For Figure 5E and F, we labeled “Intron retention with or without pSer2 Pol II gained” for the figure title. We also put the full gel images of Figure 4H in the supplementary data to show where in the gel the SDMA band migrates. For Figure 4H, we conducted new experiments using lower concentration of pemrametostat in light of the suggestion by Reviewer #2. Instead of 500 nM, we treated the cells with 200 nM pemrametostat, which is within the IC₅₀ range in both MCF-7_RBKO and T47D_RBKO cells (49.8 to 268.5 nM). In addition to MCF-7_RBKO cells, we also validated that pemrametostat inhibited SDMA levels in FUS in T47D_RBKO cells.

Fig. 2F

Fig. 5E

Fig. 5F

Supplementary Fig. 9

3. There's considerable redundancy in the ontology and association analysis data shown. For example, in Figure 5E the proteins in "SUMO E3 ligases..." seems likely to be most or all the same proteins as "SUMOylation" on the line below. There are further multiples of "Cell cycle..." and "mRNA" in Figure 5E

and 5F. The data in the main figures is also crowded with noise and irrelevance, such as “HIV infection” proteins for MCF7 cells in Figure 5F, or “Spermatogenesis” and “Heme_metabolism” in MCF7 and T47D cells in Figure 3A. The full lists of associations would be nice for the supplement tables, but they make it difficult to find what the authors describe in their results and the text is near impossible to read. It might also convey incorrect meaning, for example the three “methyltransferase” associations in red for Figure 1C highlight an inherent flaw in ontology analysis but does not make the association 3 times more important.

Thank you for the comment. The gene ontology analysis reports a list of pathways ranked by their statistical significance. Therefore, parent terms (e.g., Cell Cycle) and their child terms (e.g., Cell Cycle Mitotic) can be enriched simultaneously as top significant pathways. Although we agree with your comment, we respectfully ask for reporting the top enriched pathways without manually deletion of any parent and/or child terms to avoid introducing bias to the analysis. We are sorry for the small text in the figures and have enlarged the font size in Figure 5E and 5F according to your suggestion.

4. It’s difficult to see the change described in the text and caused by siPRMT from the stacked plot in Figure 3D. For this instance of the cell cycle data, a bar plot and significance indicated. Other stacked bar plots in the manuscript seem fine.

Thank you for the comment. We have replaced the stacked plots with bar charts and statistical significance labeled for all the cell cycle analysis (e.g., **Figs. 3D,3E,4G** and **Supplementary Figs 5,6**).

Fig. 3D

Fig. 3E

Fig. 4G

Supplementary Fig. 5

Supplementary Fig. 6

5. The bottom line about FUS seems similar to the earlier reports the authors cite demonstrating PRMT1 effects on SR and hnRNP protein activities (PMID: 31611688, 33782401, 29946150). In fact, previous studies find evidence that FUS, hnRNP proteins, and SR proteins bind each other and are recruited together to the initiating and elongating RNA polymerase (PMID: 9774382, 2285529, 26124092). However, Figure 5J implies more than this or other studies have shown. The authors imply intron inclusions, like those shown in 5H, could explain reduced protein, Figure 5I. How many reduced proteins

and increased introns correlate is measurable by whole cell proteomics, showing whether Figure 5I changes are the only options to affect cell cycle or among many scores of proteins similarly diminished. An example of one alternative model is that widespread stalling of RNA Pol II can interfere with both splicing and cell cycle, but in parallel. The same cell cycle genes of interest aren't known to be affected the same way by PRMT5i and loss of FUS. The SDMA PTM scan uncovered more RNA processing factors closely related to FUS. More scrutiny of FUS activity does not improve this study. The data is strong that targeting FUS activity validates and supports specificity of PRMT5i and FUS activity must overlap a significant portion of the mechanism of PRMT5i. It raises excitement for more study in the future of FUS and whether its activity on cell cycle can be linked to a collection of introns affected.

Thank you for the comment. As you advised, mechanisms other than intron retention can contribute to PRMT5i-mediated protein downregulation. We did not perform whole cell proteomics upon inhibition of PRMT5, so we cannot rule out other mechanisms such as RNA Pol II stalling that can affect expression of cell cycle-related genes and downregulate protein levels. We appreciate your comment and have revised our **Discussion** accordingly. Line 357-363: *"In addition to RNA splicing factors, FUS is also associated with TFIID transcription complex, suggesting that FUS may play a role in initiation of Pol II-dependent transcription [48]. Furthermore, FUS binds to Pol II on alternative poly-adenylated sites to regulate transcription termination [36]. These studies suggest that dissociation of FUS from Pol II can result in direct dysregulation of gene expression. Therefore, further studies are needed to investigate whether PRMT5i-mediated dissociation between FUS and Pol II leads to mRNA downregulation of cell cycle regulators."*

6. The inset table in Figure 6A is unclear. Why there is different shading is not explained. These have labels in the figure. It would be interesting if the authors had tested ERE luciferase activity in the presence of PRMT5i.

We apologize for the confusion. The shading in the tables in Figure 6A was generated by Excel automatically. We have removed the shading to avoid any confusion.

Fig. 6A

In response to your comment, we performed an ERE luciferase reporter assay in the presence of PRMT5i. Briefly, we transfected MCF-7_RBKO and T47D_RBKO cells with an ERE firefly luciferase reporter plasmid and a renilla luciferase control. Twenty-four hours after transfection, we treated the cells with 20-500 nM pemrametostat (Pem), 10 nM fulvestrant (Fulv), or estrogen-free media (E2-) ±1 nM 17β-estradiol (E2+) to rescue from estrogen depletion. The dual-luciferase reporter assay was conducted 48 hours after

treatment. As shown in the following figures, acute inhibition of PRMT5 did not directly modulate ER α -mediated luciferase activity. In addition, we also performed Co-IP and reciprocal Co-IP using a PRMT5 antibody and an ER α antibody, respectively. No interaction between PRMT5 and ER α was observed. The results of the ERE reporter assay and Co-IP experiment suggest that PRMT5 does not directly modulate ER α activity.

Reviewer #2 (Remarks to the Author):

In this paper the authors conclude that there is a potential role of PRMT5 inhibition to treat ER+ breast cancer patients who have failed to respond to CDK4/6 inhibitors due to loss of the RB protein. This represents a small proportion of metastatic ER+ breast cancers, although they are undoubtedly difficult to treat. The molecular findings regarding PRMT5 function are interesting. However there are multiple weaknesses in this paper described in the points below, many of which limit the observations to the models presented, rather than demonstrating real applicability in the disease setting that the authors are proposing.

1. Application to CDK4/6 inhibitor resistant ER+ breast cancer and selection of models

The authors conclude that “Collectively, these data support PRMT5 as a therapeutically actionable vulnerability to overcome resistance to CDK4/6 inhibitors in ER+/RB-deficient breast cancer.”

The authors use two main models, MCF-7 and T-47D cells, and a smattering of other models. Figures 4 and 5 exclusively describe MCF-7 cells. The models do not represent genuine CDK4/6 inhibitor resistance models that have arisen due to exposure to CDK4/6 inhibitors, but instead knockout RB models. The authors have used only one model with acquired CDK4/6 resistance (the PDX model in Figure 6), and that model is poorly described (see below). While the conclusion that RB KO ER+ cells are susceptible to PRMT5 inhibitors may be valid, the conclusion that CDK4/6 inhibitor resistant ER+ breast cancer with RB deficiency can be treated with PRMT5 inhibitors has not been shown.

This is an important distinction as there is no evidence that PRMT5 expression is maintained in RB-deficient ER+ breast cancer with resistance to CDK4/6 inhibitors. The authors could address this problem by (i) supplementing with genuine resistance models (in vitro and PDX) and (ii) examining the expression of PRMT5 in RB deleted ER+ metastatic breast cancers that are resistant to CDK4/6 inhibitors, preferably a patient cohort.

Thank you for the comment. To examine whether PRMT5 expression is maintained in ER+/RB-deficient breast cancer, we analyzed two public datasets in cBioPortal (e.g., The Metastatic Breast Cancer Project and METABRIC) and found that neither *RB1* CNV nor mutations are associated with downregulation of PRMT5 mRNA expression in ER+ breast cancer.

The Metastatic Breast Cancer Project (Provisional, Dec 2021)

METABRIC (Nature 2012 & Nat Commun 2016)

We understand that CDK4/6i-resistant models established by long-term treatment of CDK4/6i can be powerful for studying mechanisms of resistance. However, by using this approach, multiple mechanisms of resistance to CDK4/6i (e.g., *FAT1* loss, *PTEN* loss, *RAS* mutations, *CCNE1* overexpression, ...etc.) may arise and co-exist in one pooled population, which can confound interpretation of the CRISPR screening results and the identification of therapeutic targets against RB-deficient cells. Another risk is that we may not be able to recapitulate *RB1* loss-of-function alterations in the long-term treatment models if other mechanisms of resistance dominate cell growth. Therefore, we adopted the approach of reverse genetics to knockout *RB1* and then validated that RBKO drives resistance to CDK4/6i. Although we used the RBKO models in the majority of this study for proof-of-concept, we also validated our proposed combination strategy (fulvestrant plus pemrametostat) in a clinically relevant ER+/*RB1*-deleted PDX model (progressed on letrozole plus palbociclib). We apologize for the confusion regarding the description of this PDX model in our manuscript. Please see below for our point-by-point response to your questions regarding this PDX model. We appreciate your comments and have revised our manuscript accordingly to clarify the information and fidelity of this PDX model.

Particular issues with models are:

(A) The PDX model used in Figure 6D is not described adequately.

The current description is “RB1-deleted patient-derived xenografts (PDX) derived from a patient with ER+ MBC that progressed clinically on palbociclib plus the aromatase inhibitor letrozole. RB1 deletion in the PDX was confirmed by exome sequencing (data not shown)”.

- Where does this PDX come from? What human ethics approvals support its development and use?

We have added the information of the PDX in **Materials and Methods, Xenograft studies**, Line 575-578: “For the PDX model, the de-identified tissue biopsy from a patient with acquired resistance to letrozole + palbociclib was collected under the study protocol BRE03103 (<https://clinicaltrials.gov/study/NCT00899301>) with IRB approval number 030747 at Vanderbilt University.”

- How long was the patient treated for? Does this represent intrinsic or acquired resistance?

The patient was treated with letrozole plus palbociclib for 14 months and then acquired resistance to the treatment. Biopsy was taken from a liver metastasis which progressed on letrozole plus palbociclib.

- Is this PDX model actually resistant to palbociclib? The implication is that this model represents palbociclib resistance, but it is not shown in the data. PDX models often behave differently to how the disease responds in the patient.

We have characterized and confirmed that the PDX model is resistant to palbociclib (Palbo) and abemaciclib (Abema) *in vivo*. Briefly, tumor fragments were implanted into female NSG mice one day after implantation of controlled-release estradiol pellets. The mice were treated with vehicle, Palbo, or Abema daily by oral gavage when tumors reached size of 200 mm³. Please see the figure below.

- Was the RB1 deletion in the patient or only in the PDX?

RB1 deletion was confirmed by Foundation One biomarker testing. We have added the information of the PDX in **Materials and Methods, Xenograft studies**. Line 578-579: “Deletion of RB1 in the biopsy was reported by Foundation One biomarker testing.”

- Exome sequencing of the RB1 deletion by “data not shown” is inadequate.

Thank you for the comment. We have added copy number analysis in **Materials and Methods, Xenograft studies**. Line 306: “RB1 deletion in the PDX was confirmed by exome sequencing (**Materials and Methods**), ...” Line 580-584: “Genomic DNA extracted from the PDX was subjected to whole-exome sequencing at UTSW Sequencing Core. For copy number analysis, we randomly selected 10 whole-exome sequenced samples from normal breast tissue in TCGA to serve as a synthetic reference for germline alterations. Deletion of RB1 in the PDX was confirmed by comparing RB1 copy number of the PDX to that of the synthetic reference.”

(B) Cell line model selection.

The manuscript alternates between different models, and many of the findings are in one cell line only. For example the rescue experiments are T47D cells only (Figure 2D), and the mass spec/ChIP-seq is

performed exclusively in MCF7 (Figure 4/5). The FUS and Pol II data needs validation in T-47D RBKO or another model (or even better, show high FUS/activity in samples from RB KO resistant patients), as otherwise this third of the paper (Fig4/5) relies upon a single cell line. The authors generalize the FUS/Pol II data to all ER+ breast cancer, but this isn't currently supported.

Thank you for the comment. We validated the FUS/Pol II interaction in T47D_RBKO cells according to your suggestion. In Figure 4E-G, we validated that silencing of FUS results in inhibition of cell growth and G1-to-S transition in T47D_RBKO cells. In Figure 4H and I, we validated that treatment of T47D_RBKO cells with 200 nM pemrametostat blocks the interaction between FUS and Pol II. To improve the quality of the immunoblot analysis post Co-IP, we used another PRMT5 antibody (Abcam, EPR5772) that was also used for Co-IP MS. In Figure 5I and J, we validated that treatment of T47D_RBKO cells with pemrametostat results in downregulation of proteins that regulate synthesis of DNA, except for POLE in T47D_RBKO cells.

Fig. 4E

Fig. 4F

Fig. 4G

Fig. 4H

Fig. 4I

Fig. 5I

Fig. 5J

2. RB deletion versus RB deficiency

The authors have conflated the concepts of RB deficient and RB deleted in their paper. RB deletion and truncating/inactivating mutations are relatively rare in metastatic ER+ breast cancer, occurring in 3-4% of cases. The current evidence suggests that they are not acquired at high frequency in patients treated with combination endocrine therapy and CDK4/6 inhibitors. See for example “RB1 mutations arise following treatment with CDK4/6 inhibition, but .. these mutations are likely subclonal and of relatively low prevalence, suggesting, in contrast to previous work, that they are not a major mechanism of resistance.” doi: 10.1158/2159-8290.CD-18-0264

Heterozygous loss (<https://doi.org/10.1038/s41467-022-32828-6>) forms part of the RB deficient phenotype and many patients may have low RB or low RB function rather than complete loss.

The authors’ work exclusively models complete ablation of RB function by genomic deletion. As such they are describing a genotype affecting a small proportion of patients. However their language implies a much larger group of patients by using the description “RB1 deficiency”. The evidence presented in the paper does not support that PRMT5 inhibition is useful in a context of low RB function, but only that it may be useful in the context of complete RB knockout. To support these conclusions regarding RB deficiency the authors need to show that their work applies in conditions of low RB function.

Thank you for the comment. We never implied RB loss is a major mechanism of CDK4/6i resistance. But it not arguably one accepted mechanism of resistance and perhaps the one with validation in the clinic, albeit still in few reports. We are sorry for having inadvertently conflated RB ‘loss’ with RB ‘deficiency’. By the latter we meant RB loss, in line with the experimental model we used for the CRISPR screen, specifically the two RBKO cell lines.

To mimic low *RB1* expression in ER+ breast cancer, we used shRNA to silence expression of *RB1* in ER+ breast cancer cell lines HCC1428 and ZR-75-1. In comparison to the control shRNA (shGFP), shRNA-mediated silencing of *RB1* resulted in resistance to palbociclib *in vitro*. Next, we treated these cells with a dose range of pemrametostat for 6 days and then performed CyQuant analysis. Treatment of the RB-low HCC1428 and ZR-75-1 cells with pemrametostat resulted in growth inhibition with sub-micromolar IC_{50} , similar to that in MCF-7_RBKO and T47D_RBKO cells.

Supplementary Fig. 3

We appreciate your comment and have added these results in Line 152-158: “Furthermore, we tested whether ER+ breast cancer cells with low RB expression are sensitive to inhibition of PRMT5. We used shRNA to silence expression of RB1 in HCC1428 and ZR-75-1 ER+ breast cancer cells. In comparison to the control shRNA targeting GFP (shGFP), shRB1-mediated downregulation of RB resulted in resistance to the CDK4/6i palbociclib. Treatment of the RB-low HCC1428 and ZR-75-1 cells with a dose range of pemrametostat resulted in growth inhibition with sub-micromolar IC₅₀, similar to that in MCF-7_RBKO and T47D_RBKO cells (Supplementary Fig. 3).”

We also added the information related to the shRNA plasmids in **Materials and Methods**. Line 446-448: “MISSION® pLKO.1-puro shRB1 plasmid (TRCN0000288710) and GFP shRNA control plasmid were purchased from Millipore-Sigma.”

3. Some CDK4/6i resistant cancers present with downregulation of ER or a basal-like phenotype which is not hormone sensitive (and RB loss can occur in this setting). The models presented by the authors are universally ER+ and responsive to endocrine therapies with the exception of the PDX model. PRMT5 is suggested to act as an oncogene in endometrioid adenocarcinoma via the action of estrogen receptor <https://doi.org/10.1002/cjp2.194>. Are PRMT5 inhibitors going to be effective in CDK4/6i resistant cancers that have lost their reliance on ER?

We did not address that question specifically. However, data below in response to Comment #6 from Reviewer #1 suggests the effect of PRMT5i in cells with RB loss has no relationship to a reliance of those cells to ER α . Briefly, we transfected MCF-7_RBKO and T47D_RBKO cells with an ERE firefly luciferase reporter plasmid and a renilla luciferase control. Twenty-four hours after transfection, we treated the cells with 20-500 nM pemrametostat (Pem), 10 nM fulvestrant (Fulv), or estrogen-free media (E2-) \pm 1 nM 17 β -estradiol (E2+) to rescue from estrogen depletion. The dual-luciferase reporter assay was conducted

48 hours after treatment. The results of the ERE luciferase reporter assay suggest that acute inhibition PRMT5 does not directly modulate ER α activity. In addition, we performed Co-IP and reciprocal Co-IP using a PRMT5 antibody and an ER α antibody, respectively. We did not observe interactions between PRMT5 and ER α .

Additionally, we reported that inhibition of PRMT5 impedes G1-to-S cell cycle progression and suppresses growth of cancer cells (e.g., TNBC and lung cancer) that do not rely on ER signaling (**Supplementary Fig. 5**). These results suggest that inhibition of PRMT5 is effective irrespective of ER status and/or signaling.

4. Line 98-100 “CCND1 (β WT = -1.93; β RBKO = -0.23) and CDK4 (β WT = -1.75; β RBKO = -0.67) were not essential in RBKO compared to WT cells, consistent with the notion that loss of RB1 uncouples the CDK4/Cyclin D1 complex from E2F-regulated transcription and the G1-to-S transition” What is the definition of essentiality? This needs to be defined and justified.

We used MAGeCK for the analysis of the CRISPR screen. This algorithm “calculates a ‘beta score’ for each targeted gene to measure the degree of selection upon gene perturbation, similar to the ‘log fold-change’ measurement in differential expression analysis” (Nature Protocols 2019, **14**: 756-780; <https://pubmed.ncbi.nlm.nih.gov/30710114/>). In Line 96-97, we specified “The β -score represents the degree of sgRNA depletion or enrichment, with essential genes having a more negative β -score.” In Line 94-96, we specified adoption of the criterion “FDR < 0.05 and β -score < -0.5” as significant essential genes and cited a reference published by Jaselsohn R *et al.* (Cancer Cell 2018, **33**:173-186; <https://pubmed.ncbi.nlm.nih.gov/29438694/>) who used the same algorithm (MAGeCK) for their analysis of CRISPR screen.

5. Figure 2B is confusing for the sgPRMT5 MCF7 and T47D cells. How were these cell lines derived and plated for the experiment if they are unable to proliferate, as shown in Figure 2B?

We were unable to maintain sgPRMT5 transduced MCF-7 and T47D cells because of the growth arrest upon PRMT5 depletion. Therefore, sgPRMT5 MCF-7 and T47D cells needed to be freshly transduced and prepared before each experiment (e.g., growth assays and immunoblot analysis). For immunoblot analysis (**Fig. 2A**), we transduced the cells with sgPRMT5 virus and then pulled cell lysates from multiple 6-well plates right after puromycin selection. For cell growth assay (**Fig. 2B**), we seeded sgPRMT5-transduced cells into 6-well plates right after puromycin selection.

6. Figure 2F/Supp Fig 2 – what is the sensitivity of RB expressing cells to the PRMT5 inhibitor? Since the

CRISPR screen shows that RB wildtype cells are less reliant on PRMT5, a reduced sensitivity should be apparent with dose response assays.

Although the CRISPR screen shows that sgPRMT5 had a lower β -score in RBKO cells (-2.76) than in RB expressing parental cells (-1.51), PRMT5 appears to be an essential gene in both RB1 genotypes as their sgPRMT5 dropped out significantly (FDR <0.05 and β -score <-0.5). We did the dose response assays and found no differences in the sensitivity to PRMT5i between RBKO and RB WT cells. Although inhibition of PRMT5 is not synthetic lethal to ER+/RB-deficient breast cancers, targeting PRMT5 suppresses G1-to-S cell cycle progression in the absence of RB and thus serves as a potential therapeutic strategy to treat this genotype of breast cancer. We have added the results of WT cells and revised our **Discussion** accordingly. Line 381-391: *“In our study herein, we demonstrate that both SAM-cooperative (e.g., pemrametostat) and SAM-competitive (e.g., JNJ-64619178) PRMT5i suppress growth of both ER+/RB-deficient and RB-competent breast cancers in vitro and/or in vivo. Inhibition of PRMT5 induces intron retention and subsequently downregulates corresponding proteins that drive DNA synthesis in the S phase. Therefore, inhibition of PRMT5 bypasses the CDK4/6-RB-E2F regulatory axis and thus impedes G1-to-S transition independent of RB. Although targeting PRMT5 is not synthetic lethal to RB-deficiency, its mechanism of cell cycle inhibition provides a rationale for future studies testing first-generation PRMT5i also in RB-competent, CDK4/6i-refractory breast cancers with other mechanisms of resistance (e.g., CCNE1 overexpression, FAT1 loss, PTEN loss, etc.) [9, 11, 56, 57].”*

Fig. 2F

Supplementary Fig. 2

7. All cell cycle data needs statistical analysis. Why don't all the cell cycle distribution experiments add up to 100%? Where is the gating strategy data?

Thank you for the comment. We have added statistical analysis for the cell cycle data and changed the stacked plots to individual bar charts for better visualization (e.g., **Figs. 3D,3E,4G** and **Supplementary Figs**

5,6). For the cell cycle analysis, we used univariate modeling in FlowJo: “To preserve the relative relationship between phases we do not simply normalize to 100% but rather map the probabilities to those that sum to 100%. This process can sometimes lead to percentages that add up to a little more or a little less than 100%, or small negative numbers, particularly in the sub-G0/G1 or super G2/M populations.” – FlowJo, Univariate Statistics (<https://docs.flowjo.com/flowjo/experiment-based-platforms/cell-cycle-univariate/plat-cc-statistics/>).

Fig. 3D

Fig. 3E

Fig. 4G

Supplementary Fig. 5

Supplementary Fig. 6

We removed doublets and debris by gating FSC and SSC and then performed cell cycle analysis using the FlowJo algorithm. Since the cells were stained with PI only, no compensation was needed. Please see below for examples of the gating strategy.

8. Fig 4 – The authors assert that FUS knockdown phenocopies PRMT5 knockdown. This would be more convincing with some sort of rescue experiment. For example, does FUS knockdown cause cell cycle effects in cells with re-expressed WT PRMT5, but not in PRMT5 E444Q cells (as per Figure 2D)?

Thank you for the comment. We agree that a rescue experiment can provide more evidence to demonstrate that FUS is a downstream effector of PRMT5. Our hypothesis is that inhibition of PRMT5 disrupts interaction between FUS and Pol II. Therefore, a rescue experiment should be performed by expressing a PRMT5-independent version of FUS. Theoretically, the PRMT5-independent FUS should bind

to Pol II irrespective of PRMT5 activity and should restore RNA splicing dysregulated by PRMT5i. However, to the best of our knowledge, there are no known FUS mutations that result in constitutive binding to Pol II. Additionally, it is unclear whether FUS-mediated RNA splicing remains unaffected when FUS constitutively binds to Pol II. Mutations in arginine or truncations of RGG/RG domain in FUS may be detrimental because SDMA level is positively correlated to the interaction between FUS and Pol II (**Fig. 4H**). Although we understand that rescue experiments could provide more evidence for proof-of-concept, we are limited by models that are available for this particular experiment. In alternative, we show that 1) *FUS* knockdown resulted in inhibition of G1-to-S transition as did *PRMT5* knockdown (**Fig. 4E-G**), and 2) PRMT5i resulted in elevation of pSer2 Pol II and dysregulation of RNA splicing (**Fig. 5**), which is consistent with *FUS* knockdown as reported previously (*Gene Dev* 2012, **26**: 2690-2695; <https://pubmed.ncbi.nlm.nih.gov/23249733/>). We believe that these data are consistent with the conclusion that knockdown of *FUS* phenocopies inhibition of PRMT5.

9. HCI-018 is used in this paper, but no evidence is presented that it is RB-deleted. This is needed. Also, is this model resistant to palbociclib?

Thank you for the comment. HCI-018 was kindly provided by Dr. Alana Welm. In Dr. Welm's publication (*Nature Cancer* 2022, **3**: 232-250; <https://www.nature.com/articles/s43018-022-00337-6>), HCI-018 was sequenced and identified as *RB1* CNV-low. Although we did not sequence again HCI-018, we confirmed loss of RB protein expression in HCI-018 by IHC (please see below). We also validated that treatment with palbociclib failed to inhibit growth of HCI-018 in 3D culture (**Fig. 2H**).

10. Figure 5I: Are these proteins downregulated due to pemrametostat treatment, or due to off-target effects? Downregulation is only convincing at the high dose in Fig 5I (500nM, which is 3x the IC50), but not at the low dose, which raises the likelihood of off-target effects. Are these same proteins downregulated with PRMT5 shRNA or siRNA? I would find this whole figure more convincing if intron retention was also shown with PRMT5 knockdown.

Thank you for the comment. In light of your suggestion, we performed immunoblot analysis using lysates of MCF-7_RBKO and T47D_RBKO cells treated with pemrametostat at 0, 20, 100, and 500 nM. We observed significant protein downregulations in a concentration dependent manner, except for POLE in T47D_RBKO.

Fig. 5I

Fig. 5J

11. Many other experiments are carried out with 500nM pemrametostat, and these doses also seem high, given that this dose leads to 25% cell viability after 6 days treatment. What is the overlap between the RNAseq on MCF7 RBKO treated with pemrametostat performed in Fig 5C, and PRMT5 siRNA RNAseq on MCF7 RBKO in Figure 3? If pemrametostat is being applied at a dose that mainly acts through PRMT5 then there should be excellent overlap between these datasets.

We compared the RNA-seq between MCF-7_RBKO treated with pemrametostat and MCF-7_RBKO transfected with siPRMT5 as the Reviewer suggested. Briefly, we filtered out non-expressed genes (FPKM <1) and compared significantly up- or down-regulated genes (FDR <0.05) or non-significant genes (FDR >0.05) between the two datasets using hypergeometric test. We found significant overlap between the two datasets. Please note that several factors may affect the eventual experimental results of the RNA-seq. For instance, siPRMT5 knockdown efficiency; time of onset is different between immediate pharmacological inhibition and siRNA-mediated mRNA downregulation and subsequent protein downregulation; stability of pemrametostat and siRNA may be different. Therefore, we would not be able to expect complete overlap between the two datasets.

We performed several key experiments using lower concentrations of pemrametostat to address the Reviewer's suggestion. We validated that 200 nM pemrametostat was able to inhibit interaction between FUS and Pol II in both MCF-7_RBKO and T47D_RBKO cells (Fig. 4H,I). Additionally, we validated that pemrametostat at 100 nM was able to downregulate proteins that modulate DNA synthesis (Fig. 5I,J).

Fig. 4H

Fig. 4I

Fig. 5I

Fig. 5J

12. No uncropped western blots are shown. Densitometry of replicate western blots experiments are lacking eg 2C, 2F, 5I.

Thank you for the comment. We have added uncropped Western blots in **Supplementary Fig. 9**. We have also performed quantitative analysis according to the Reviewer's suggestion.

Reviewer #3 (Remarks to the Author):

The paper by Lin et al describes the discovery of the protein arginine methyltransferase PRMT5 as a potential therapeutic target in CDK4/6 inhibitor-resistant breast cancer that are ER+/RB-deficient. This well-structure and composite paper addresses an important clinical issue, provides convincing mechanistic insights on PRMT5 activity in the tumor subtype under investigation and is very well written. The findings are largely compelling and appropriate for the wide audience of the journal. Therefore I recommend publication of the manuscript after minor revisions.

Few minor concerns should be considered prior to publication:

- 1) The authors performed a genome-wide CRISPR screen on both WT and RBKO cells and defined genes essential for one or both conditions. However, PRMT5 is essential for both. While there is no doubt that PRMT5 is a valuable target in RBKO, the authors should better discuss its potential role in WT cells.

Thank you for the comment. We also treated WT cells with pemrametostat and another PRMT5i JNJ64619178. There were no differences between WT and RBKO cells in sensitivity to these PRMT5i. We appreciate your suggestion and have revised our **Results** and **Discussion** accordingly. In **Results**, Line 144-146: “However, there were no differences between WT and RBKO cells in sensitivity to pharmacological inhibition of PRMT5 (Fig. 2F and Supplementary Fig. 2).” In **Discussion**, Line 381-391: In our study herein, we demonstrate that both SAM-cooperative (e.g., pemrametostat) and SAM-competitive (e.g., JNJ-64619178) PRMT5i suppress growth of both ER+/RB-deficient and RB-competent breast cancer in vitro and/or in vivo. Inhibition of PRMT5 induces intron retention and subsequently downregulates corresponding proteins that drive DNA synthesis in the S phase. Therefore, inhibition of PRMT5 bypasses the CDK4/6-RB-E2F regulatory axis and thus impedes G1-to-S transition independent of RB. Although targeting PRMT5 is not synthetic lethal to RB-deficiency, its mechanism of cell cycle inhibition provides a rationale for future studies testing first-generation PRMT5i also in RB-competent, CDK4/6i-refractory breast cancers with other mechanisms of resistance (e.g., CCNE1 overexpression, FAT1 loss, PTEN loss, etc) [9, 11, 56, 57].”

Fig. 2F

Supplementary Fig. 2

2) In Figure 1D PRMT1 is also highlighted, but is not cited in the text: since PRMT1 is the most active type-I PRMT, with wide range of non-histonic protein targets and experimentally-reported cross-talk with PRMT5, it would be important to discuss this piece of evidence in the context of the story presented.

We appreciate this suggestion and have revised our **Discussion** accordingly. Line 364-371: “Recent studies have demonstrated compensatory crosstalk between PRMT1 and PRMT5, and combined inhibition of PRMT1 and PRMT5 leads to synergistic antitumor effects [40, 49, 50]. Indeed, FUS is also a substrate for

PRMT1 [51]. Although we did not examine levels of MMA and ADMA in FUS or in tumors in vivo, it is possible that other PRMT family proteins can partially compensate pemrametostat-mediated SDMA suppression in FUS. Of note, our CRISPR screen also identified Type I PRMTs (e.g., PRMT1 and CARM1) as essential genes in RBKO cells. Therefore, further investigation for dual inhibition of PRMT1 and PRMT5 is warranted.”

3) more information regarding the number of technical/biological replicates (for instance for western blots experiments) should be provided in the figure legends.

Thank you for the comment. We have added the number of biological replicates for quantitative Western blot experiments in the figure legends (e.g., **Figs. 2C,2E,4H,5I**).

4) the statistical significance should be added in Figs 2B, 3D, 3E, 4G, 6B, 6D, S4C, S5C.

Thank you for the comment. We have added statistical significance in **Figs 2B, 3D, 3E, 4G, 6B, 6D, S5C, S6C** and revised figure legends accordingly. In Figure 2B, we observed statistical growth inhibition in MCF-7_WT cells transduced with sgPRMT5. We apologize for not interpreting the data accurately in the original submission by reading the growth pattern without statistical analysis. We have corrected our statement in **Results**. Line 124-127: *“Consistent with the CRISPR screening results, PRMT5 depletion resulted in statistical growth inhibition in isogenic WT and RBKO of both MCF-7 and T47D cells, except for one of the sgPRMT5 in MCF-7_WT cells (Fig. 2B).”*

Fig. 2B

5) the x axis legend should be added in Fig. 6B-D (right plot)

We have labeled the x axis in Fig. 6B,D.

Fig. 6B

Fig. 6D

6) the list of co-enriched proteins in panel 4A and of R-methylated peptides down-regulated upon siPRMT5 in figure 4B should be reported (besides uploading the MS raw data in open access repository), to gain a more complete view of the quantitative proteomic data acquired. The volcano-plot displayed in panel 4B (with absolutely not a single di-methylated peptide that are upregulated upon siPRMT5) is a bit surprising, in light of the well-documented scavenging elicited through PRMT1 when PRMT5 is inactivated). all in all, the proteomics data are presented in a too simplistic manner to the fully convincing: more data (in terms of txts data output derived by the quantitative analysis of MS raw data and statistical analysis applied) should be provided.

Thank you for the comment. We have provided the quantitative and statistical analysis of MS data in Supplementary Tables 2,3. PRMT1 is a type I PRMT which drives formation of ADMA. Therefore, the SDMA PTM analysis shown in Figure 4B may not capture changes in ADMA levels.

Thank you again for the constructive critique of our study. We hope this rebuttal addresses your concerns and that our paper is reconsidered for publication in *Nature Communications*.

Reviewers' Comments:

Reviewer #1:

Remarks to the Author:

The authors have carefully responded and provided new data that satisfy all my suggestions made on the first submission. I recommend the manuscript for publication in Nature Communications.

Reviewer #2:

Remarks to the Author:

Thankyou to the authors for providing comprehensive responses to my queries, and to those of the other reviewers. My queries have been either answered to my satisfaction, or good explanations provided as to why available resources do not permit a thorough exploration of my queries.

Supplementary western blots have now been provided, but these are lacking ladders/MW markers. These are necessary for interpretation of data, especially as some membranes have been cut.

Reviewer #3:

Remarks to the Author:

The authors have satisfactorily addressed the criticisms raised by the reviewers and this revised version of the manuscript is further improved in terms of clarity, consistency and impact of the findings. Hence I think that the revised manuscript is worth publications in Nat. Comms.